# Survival of the simplest in microbial evolution

Torsten Held [1,2], Daniel Klemmer[1,2] & Michael Lässig[1]

The evolution of microbial and viral organisms often generates clonal interference, a mode of competition between genetic clades within a population. Here we show how interference impacts systems biology by constraining genetic and phenotypic complexity. Our analysis uses biophysically grounded evolutionary models for molecular phenotypes, such as fold stability and enzymatic activity of genes. We find a generic mode of phenotypic interference that couples the function of individual genes and the population's global evolutionary dynamics. Biological implications of phenotypic interference include rapid collateral system degradation in adaptation experiments and long-term selection against genome complexity: each additional gene carries a cost proportional to the total number of genes. Recombination above a threshold rate can eliminate this cost, which establishes a universal, biophysically grounded scenario for the evolution of sex. In a broader context, our analysis suggests that the systems biology of microbes is strongly intertwined with their mode of evolution.

[1] Institut für Biologische Physik, Universität zu Köln, Zülpicherstr. 77, 50937 Köln, Germany. [2] These authors contributed equally: Torsten Held, Daniel Klemmer. Correspondence and requests for materials should be addressed to M.L. (email: mlaessig@uni-koeln.de)

In the absence of recombination, evolution is constrained by genetic linkage. That is, selection on an allele at one genomic locus can interfere with the evolution of simultaneously present alleles throughout the genome. Interference interactions between loci include background selection (the spread of a beneficial allele is impeded by linked deleterious alleles), hitchhiking or genetic draft (a neutral or deleterious allele is driven to fixation by a linked beneficial allele), and clonal interference between beneficial alleles originating in disjoint genetic clades (only one of which can reach fixation). These interactions and their consequences for genome evolution have been studied extensively in laboratory experiments[1,2] and in natural populations[3,4]. Recent theory[5–13] has quantified two broad interference effects in asexual evolution. First, interference selection rather than genetic drift constrains the genetic diversity in large populations, which, in turn, limits the efficacy of selection[10,13–15]. Second, interference reduces the speed of evolution[7–9,11–13]; this has been observed in laboratory evolution experiments[16–19]. The resulting fitness cost of interference, which has also been been observed in microbial laboratory evolution[20–23], is the center piece of classic arguments for the evolutionary advantage of sex[24–28].

Much less clear is how interference affects the evolution of molecular phenotypes, such as protein stabilities and affinities governing gene regulation and cellular metabolism. The systems-biological consequences of interference evolution are the topic of this paper. Our analysis is based on biophysical models of molecular evolution[29–36]. In a minimal model, each gene of an organism carries a single quantitative trait $G$, the stability of its protein fold. A fitness landscape $f(G)$ quantifies the effect of protein stability on reproductive success. This landscape is a sigmoid function with a high-fitness plateau corresponding to stable proteins and a low-fitness plateau corresponding to unfolded proteins (Fig. 1a). We also discuss a stability-affinity protein model with a two-dimensional fitness landscape $f(G, E)$; this model includes enzymatic or regulatory functions of genes, specifically the protein binding affinity $E$ to a molecular target. From the perspective of molecular evolution, these landscapes provide a generic biophysical model of local fitness epistasis, which couples all sequence sites contributing to a stability or affinity trait in the same gene. Importantly, local epistasis in

protein-coding sequence operates independently of fitness interactions across genes. Beyond proteins, local epistasis occurs ubiquitously in quantitative molecular traits associated with binding interactions. This form of epistasis is an important building block of our model that is not covered by the standard theory of asexual evolution[5–13].

The system-wide evolution of molecular quantitative traits under genetic linkage defines a particular mode of phenotypic interference, which occurs broadly under conditions of typical microbial systems. This mode couples global and local evolution in a specific way: the global pace of evolution sets the average selection coefficient of local trait changes. In the first part of the paper, we develop the theory of phenotypic interference and derive a key quantitative result: in a system of $g$ genes, the steady-state fitness cost of interference increases quadratically with $g$. This super-linear cost reflects a specific evolutionary mechanism: each additional gene degrades stability and function of all other genes by increasing the accumulation of deleterious mutations. We then turn to biological implications of phenotypic interference. We show that the interference cost can outweigh the metabolic cost of genes[37,38] and generate long-term impact on systems biology: it strongly constrains genome complexity in viable, asexually reproducing organisms and drives the loss of non-essential genes. On the time scales of laboratory evolution experiments, phenotypic interference reduces fitness through the attrition of molecular traits; we compare this prediction to experimental data[20–23]. Finally, phenotypic interference provides a surprisingly simple pathway for the evolution of sex. We show that facultative recombination at low rates $R$ can evolve near neutrality yet, once $R$ exceeds a threshold $R^\star$, provides a large competitive advantage against competing non-recombining lineages. The predicted threshold $R^\star$ is of order of the mutation rate, which is consistent with observed recombination rates.

## Results

**Housekeeping evolution under phenotypic interference.** Here we analyze the evolution of genetically linked systems in a conservative environment, where populations maintain the functionality of molecular traits in the presence of deleterious

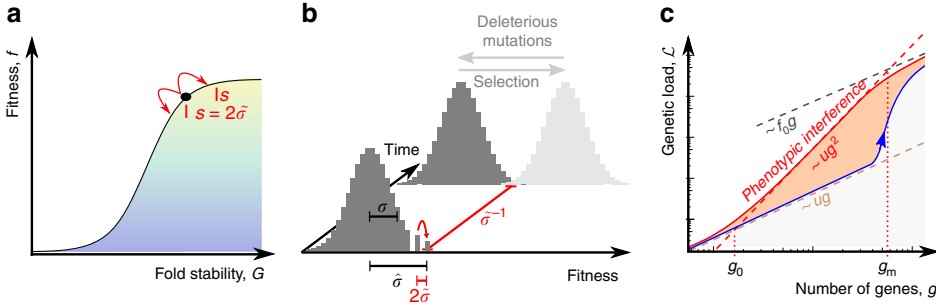

**Fig. 1** Phenotypic interference. Phenotypic interference is an evolutionary mode of multiple quantitative traits under genetic linkage with the following features: **a** Local evolution. Individual traits $G$ change by mutations (red arrows) with mean square selection coefficient $s^2$ set by the slope of the fitness landscape $f(G)$. In housekeeping equilibrium, the traits have detailed balance between beneficial and deleterious substitutions[30]. **b** Global evolution proceeds in a fitness wave, i.e., multiple genetic and phenotypic variants co-exist in a population[5,7–9,11–13]. The wave is fueled by new beneficial mutation at its tip (red arrows). It has fitness variance $\sigma^2$ and a total fitness span $\hat{\sigma} = \sqrt{c}\sigma$ with a factor $c$ depending weakly on the population size (see Methods section). By Fisher's theorem, these quantities determine the coalescence rate $\tilde{\sigma} = \sigma^2/\hat{\sigma} = \sigma/\sqrt{c}$ (see Methods section). Its inverse is the coalescence time (i.e., the average time to the most recent common ancestor of two individuals), which is proportional to the effective population size, $\tilde{\sigma}^{-1} = 2N_e$ (red bar)[13,14]. In housekeeping equilibrium, the selective advance (light gray) is offset by deleterious mutations, i.e., there is no net (local or global) change in fitness[12,77] (dark gray). Feedback between global and local evolution: the global coalescence rate sets selection on individual traits, $s^2 = 4\tilde{\sigma}^2$ (Eq. 3). The wave complexity, measured by the number of simultaneously segregating beneficial trait mutations destined for fixation, is of order $2\hat{\sigma}/s = c$ (Eq. 21). **c** Superlinear load. The genetic load $\mathcal{L}$ depends quadratically on the number of genes, $g$, over a broad range $g_0 \lesssim g \lesssim g_m$ (red line, see text). This load is substantially higher than the load under asexual evolution with discrete gene fitness effects (blue) and under of recombination (brown)

mutations but there is no adaptive pressure on these traits. This scenario defines a system-wide mutation-selection steady state that we call housekeeping evolution (here, housekeeping does not refer to a particular class of genes or metabolic processes). It builds on the assumption that over long time scales, selection acts primarily to repair the deleterious effects of mutations, because these processes are continuous and affect the entire genome. In contrast, adaptive processes are often environment-dependent and transient, and they affect only specific genes. In Methods section, we extend our analysis to scenarios of adaptive evolution and show that these do not affect the conclusions of the paper.

Figure 1 illustrates the ingredients of phenotypic interference in the housekeeping state (and can serve as a shortcut through theory for readers primarily interested in the biological implications). First, local quantitative traits of a given gene are in an evolutionary equilibrium, where the long-term average of the trait value and its position on the fitness landscape are determined by the uphill force of selection and the downhill force of mutations (Fig. 1a and Supplementary Fig. 1a). Second, global genome evolution takes place in a so-called fitness wave; that is, genetic and phenotypic variants in multiple genes co-exist in a population and generate a broad distribution of fitness values[7–9,11–13] (Fig. 1b and Supplementary Fig. 1b). These levels are linked by a common evolutionary parameter, the coalescence rate $\tilde{\sigma}$, or equivalently by the effective population size $N_e = (2\tilde{\sigma})^{-1}$ (Supplementary Table 1 lists all mathematical symbols). The joint solution of the local and global evolutionary dynamics identifies a broad regime of phenotypic interference, which is marked by a system-wide genetic load depending quadratically on genome size (Fig. 1c).

**Evolution of a quantitative trait under interference selection.** In the framework of the minimal biophysical model, we study the housekeeping evolution of genome-wide protein fold stability. The stability trait $G$ of a given gene is defined as the free energy difference between the unfolded and the folded state (and usually denoted by $\Delta G$; we abbreviate this notation to avoid confusion with the variance measures defined below). The trait $G$ evolves in a fitness landscape $f(G)$ of sigmoid form (Fig. 1a, see Methods section).

The mutation-selection equilibrium on a flank of the landscape $f(G)$ can be characterized by the equilibrium values of its population mean trait, $\Gamma \equiv \overline{G}$, and the trait diversity or genetically heritable trait variance, $\Delta_G \equiv \overline{G^2} - \Gamma^2$ (overbars denote averages within a population[39]). First, the diversity $\Delta_G$ takes the simple, effectively neutral equilibrium form

$$\Delta_G = \frac{u\epsilon_G^2}{2\tilde{\sigma}}, \tag{1}$$

which is proportional to the total mutation rate $u$ and the mean square stability effect $\epsilon_G^2$ of the relevant sequence sites, and to the effective population size $N_e = (2\tilde{\sigma})^{-1}$. This form extends previous results on neutral sequence diversity[14,40–42] and on quantitative trait diversity under genetic drift[43–45]. In Methods section, we derive Eq. 1 for quantitative traits in a fitness landscape $f(G)$ by showing that stabilizing selection on $\Delta_G$ can be neglected throughout the phenotypic interference regime; this scaling is confirmed by simulations (Supplementary Fig. 2a). In a fitness wave, the parameter $\tilde{\sigma}$ couples each individual trait to the global evolutionary dynamics of all genetically linked genes (Fig. 1a, b). In contrast, an independently evolving trait would depend on an effective population size $N_e$ set by genetic drift. Next, we compute the equilibrium point for the mean trait $\Gamma$ by equating the rate of stability increase by selection with the rate of stability degradation

by mutations,

$$\Delta_G f'(\Gamma) = u\epsilon_G; \tag{2}$$

details are given in Methods section. This mutation-selection equilibrium depends on the effective population size, in contrast to protein evolution models in the infinite population limit[31]. By inserting Eq. 1, we can express the mean square selection coefficient at trait sites, $s^2 = \epsilon_G^2 f'^2(\Gamma)$, and the fitness variance $\Delta_f \approx \Delta_G f'^2(\Gamma)$ in terms of the coalescence rate,

$$s^2 = 4\tilde{\sigma}^2, \quad \Delta_f = 2u\tilde{\sigma}; \tag{3}$$

a similar relation for $s^2$ under genetic drift has been derived in refs. [46,47]. These equations describe stable trait equilibria on the downward-curved shoulder of the fitness landscape $f(G)$, which is a non-linear trait interval with $f''(G) < 0$. They express universal characteristics of these equilibria, which do not depend on details of the fitness landscape and of the trait effect distribution of sequence sites. Their validity is confirmed by numerical simulations (Supplementary Fig. 2). The above derivation neglects fluctuations of $\Gamma$ by genetic drift and genetic draft; cf. Supplementary Fig. 1a. However, Eq. 3 remain exactly valid in the full mutation-selection-coalescence dynamics (Supplementary Methods 1 and Supplementary Fig. 3).

A salient feature of selection on quantitative traits becomes apparent from Eq. 3: the selection coefficients of new genetic variants are not fixed a priori, but are an emergent property of the global evolutionary process. A faster pace of evolution, i.e., an increase in coalescence rate $\tilde{\sigma}$, reduces the efficacy of selection[10,11,14]. On the downward curved shoulder of the fitness landscape, this drives the population to an equilibrium point of lower fitness and higher fitness gradients. In other words, trait-changing mutations are under ubiquitous negative epistasis: the combined (log) fitness effect of two deleterious trait changes is larger in magnitude, the combined effect of two beneficial mutations is smaller than the sum of the individual effects. This epistasis tunes typical selection coefficients to marginal relevance, where mean allele sojourn times between low and high frequencies, $1/s$, are of the order of the coalescence time $2N_e = 1/\tilde{\sigma}$. That point marks the crossover between effective neutrality ($s \ll \tilde{\sigma}$) and strong selection ($s \gg \tilde{\sigma}$)[10]; consistently, most but not all trait sites carry their beneficial allele.

**Interference of multiple traits.** We now obtain a closed solution of housekeeping evolution under phenotypic interference by matching the individual trait equilibria given by Eq. 3 with a fitness wave model for global evolution. First, the total fitness variance $\sigma^2$ is simply the sum of the fitness variances $\Delta_f$ of the individual genes (Supplementary Fig. 4). Using Eq. 3, this sum rule takes the form $\sigma^2 = g\Delta_f = 2ug\tilde{\sigma}$, which relates the scales of global selection and coalescence, $\sigma$ and $\tilde{\sigma}$. Second, given a sufficient supply of non-neutral mutations, global evolution proceeds in a fitness wave (the condition for wave occurrence will be made precise below). General fitness wave theory then provides another relation between global selection and coalescence,

$$c \equiv \frac{\sigma^2}{\tilde{\sigma}^2} = c_0 \log(N\sigma), \tag{4}$$

where $N$ is the population size and $c_0$ is a model-dependent prefactor[12,13] (Methods). Combining these relations, we obtain the global fitness wave of phenotypic interference,

$$\sigma^2 = 4\frac{u^2 g^2}{c}, \quad \tilde{\sigma} = 2\frac{ug}{c}. \tag{5}$$

Equations 3 then determine the corresponding characteristics of individual traits,

$$\frac{\Delta_G}{\epsilon_G^2} = \frac{c}{4g}, \quad s^2 = 4\tilde{\sigma}^2 = 16\frac{u^2 g^2}{c^2}. \tag{6}$$

Equations 5 and 6 involve the fitness wave parameter defined in Eq. 4,

$$c \simeq c_0 \log \frac{2Nug}{c_0}, \tag{7}$$

which depends only weakly on the evolutionary parameters and provides corrections to the scaling. This parameter estimates the complexity of the fitness wave, that is, the average number of genes with simultaneously segregating beneficial genetic variants destined for fixation (Fig. 1 and see Methods section). A wave pattern with temporally stable fitness polymorphism of approximately Gaussian form occurs whenever the mutation rate exceeds the average site selection coefficient, $ug \gtrsim s$ [15]. This regime underlies the closure of Eqs. 5, 6; cf. Supplementary Fig. 1b. As shown in Methods section, it applies to gene numbers above a threshold $g_0$ given by the condition

$$g_0 = \frac{c}{4}. \tag{8}$$

These relations are the centerpiece of phenotypic interference theory. They show that the collective evolution of molecular quantitative traits under genetic linkage depends strongly on the number of genes that encode these traits. The dependence is generated by a feedback between the global fitness variation, $\sigma^2$, and mean square local site selection coefficients, $s^2$. This feedback also tunes the evolutionary process to the crossover point between

independently evolving genomic sites and strongly correlated fitness waves composed of multiple small-effect mutations (Supplementary Methods 2). Remarkably, local and global characteristics of phenotypic interference are strongly universal: they depend only on the parameters $g$, $u$, and $c$ but decouple from details of gene fitness landscapes and site effect distributions.

The scaling of phenotypic interference is confirmed by extensive numerical simulations of Fisher-Wright populations, which are detailed in Methods section. Figure 2 shows the global observables $\sigma^2$, $\tilde{\sigma}^2$ and the local observables $\Delta_G$, $s^2$ as functions of $g$. The data display a crossover from a weak-interference regime of independently evolving genes at low values of $g$ (brown dashed lines) to the phenotypic interference scaling given by Eqs. 5–7 (red dashed lines); this crossover occurs around a modest gene number $g_0 \sim 100$. The calibration between theory and data involves the fitting of a single model-dependent amplitude $c_0$; the calibrated theory matches the data for realistic gene numbers ($g \sim 10^3 - 10^4$) without additional fit parameters. The data also show the universality of the leading scaling behavior; gene selection coefficients $f_0$ varying by more than three orders of magnitude introduce only small corrections to scaling. Supplementary Fig. 1 displays the separation of diversity scaling between predominantly monomorphic individual traits and standing fitness variation, as detailed in Eqs. 19, 20 of Methods section. The underlying near-linear relation between global fitness variance $\sigma^2$ and coalescence rate $\tilde{\sigma}^2$, which is a general property of fitness waves, is checked in Supplementary Fig. 2d.

**Interference selection against complexity.** The evolutionary cost of deleterious mutations is quantified by the genetic load, which is defined as the mean fitness of a population compared to the

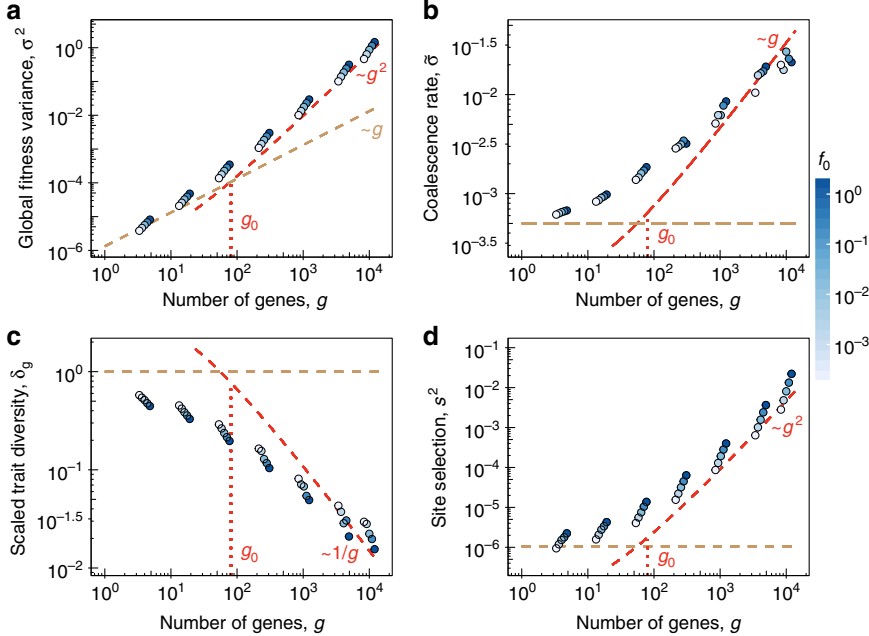

**Fig. 2** Global and local scaling under phenotypic interference. Average population data are plotted against the number of genes, $g$, for the minimal biophysical model at steady-state asexual evolution. Simulation data (circles) for different average gene selection coefficients $f_0$ (indicated by color) show a crossover from the regime of independently evolving genes (brown dashed lines) to phenotypic interference (red dashed lines). The error bars (SD) are smaller than or equal the diameter of the symbols. The crossover point $g_0 \sim 10^2$ is marked. **a, b** Global observables. Average total fitness variance, $\sigma^2$, and coalescence rate, $\tilde{\sigma}$; phenotypic interference scaling $\sigma^2 \sim g^2/c$ and $\tilde{\sigma} \sim g/c$ (red dashed lines) as given by Eqs. 5, 7 with a fit parameter $c_0 \approx 80$. **c, d** Local observables. Average scaled trait diversity $\delta_G = \Delta_G/\epsilon_G^2$ and mean square selection coefficient at sequence sites $s^2$; phenotypic interference scaling $\delta_G \sim c/g$ and $s^2 = 4\tilde{\sigma}^2 \sim g^2/c^2$ (red dashed lines) as given by Eqs. 6, 7. Values $\delta_G < 1$ indicate that individual proteins are in the low-mutation regime. The scaling $s^2 \sim \tilde{\sigma}^2$ is independent of $f_0$, signaling that site selection coefficients emerge from a feedback between global and local selection (see text). Other simulation parameters: $N = 1000$, $u = 1.25 \times 10^{-3}$, $\epsilon_G/k_B T = 1$; see Methods section for simulation details. Supplementary Fig. 2 shows global and local observables as functions of $\tilde{\sigma}$

fitness maximum. In the biophysical fitness landscape $f(G)$ of the minimal model, the load of a given gene takes the approximate form $f_0 - f(\Gamma)$, where $\Gamma$ denotes the population mean stability and $f_0$ is the fitness of a fully stable gene ($G \gg 0$); see Fig. 1a and Eq. 13 in Methods section. We now compute the genetic load under phenotypic interference for stable and functional genes, which are located in the concave part of the minimal model landscape $f(G)$. This part can be approximated by its exponential tail, where the load is proportional to the slope $\mathcal{L} = f'(\Gamma)/k_B T$. Equation 6, $s = \epsilon f'(\Gamma) = 2\tilde{\sigma}$, then predicts a load $s k_B T/\epsilon_G \approx 2\tilde{\sigma}$ per gene, where we have used that typical reduced effect sizes $\epsilon_G/k_B T$ are of order 1 (see Methods section). With Eq. 5, we obtain a quadratic scaling of the total equilibrium genetic load,

$$\mathcal{L}_{\text{int}}(g) \approx 2g\tilde{\sigma} = \frac{4ug^2}{c}, \qquad (9)$$

which sets on at a small gene number $g_0$ given by Eqs. 7, 8 (Fig. 1c; numerical simulations are shown in Fig. 3). The superlinearity of the load is the most important biological consequence of phenotypic interference and the main difference to previous results on protein evolution[31]. It is generated by the evolutionary feedback between global and local selection discussed in Fig. 1: increasing the number of genes reduces the coalescence time $\tilde{\sigma}^{-1}$ and, thus, the efficacy of selection on every single gene.

In Supplementary Methods 3 and Supplementary Fig. 5, we discuss phenotypic interference in extended biophysical models. These include active protein degradation at the cellular level, a ubiquitous process that drives the thermodynamics of folding out of equilibrium[48]. Another example is the stability-affinity model, which has two quantitative traits per gene that evolve in a two-dimensional sigmoid fitness landscape $f(G, E)$[35,49]. Under reasonable biophysical assumptions, evolution in the stability-affinity model produces a 2-fold higher interference load than the minimal model, $\mathcal{L}_{\text{int}}(g) \approx 8ug^2/c$. Alternative models with a quadratic single-peak fitness landscape describe, for example, gene expression levels under stabilizing selection[50]. Such landscapes generate an even stronger load nonlinearity, $\mathcal{L}_{\text{int}}(g) \sim g^3$. In contrast, a discrete model with a fitness effect $f_0$ of each gene shows a linear load up to a characteristic gene number $g_m = (f_0/u)\log(Nf_0)$ associated with the onset of mutational meltdown

by Muller's ratchet[8,51,52]. These examples suggest that superlinear scaling of the genetic load holds quite generally, given a sufficient number of quantitative traits evolving under genetic linkage and in fitness landscapes with negative epistasis. This type of landscape is ubiquitous in biophysical models.

The equilibrium load $\mathcal{L}_{\text{int}}$ generates strong long-term selection against genome complexity: the fitness cost for each additional gene, $\mathcal{L}'_{\text{int}}(g)$, can take sizeable values even at moderate genome size. For example, in a "standard" microbe of the complexity of E. coli, a 10% increase in gene number may incur an additional load $\Delta\mathcal{L} \approx 3 \times 10^{-2}$ under the stability-affinity model (with parameters $g = 5000$, $u = 10^{-6}$, $N = 10^8$). This estimate should be regarded as a lower bound, which is based only on core protein functions but ignores, for example, regulatory functions encoded in intergenic DNA. In comparison, the discrete model leads to a much smaller value $\Delta\mathcal{L} = 5 \times 10^{-4}$ for the same parameters.

**Genetic load can exceed metabolic fitness cost**. We can compare the interference load $\mathcal{L}'_{\text{int}}(g) = 8ug/c$ of an extra gene with its physiological fitness cost $\mathcal{L}'_{\text{phys}}(g)$, which is generated primarily by the synthesis of additional proteins (and is part of the fitness amplitude $f_0$). Metabolic theory shows that spurious expression leads to a re-allocation of metabolic resources in the cell and a reduced growth rate, $\lambda = \lambda_0(1 - \phi_U/\phi_{\max})$, where $\phi_U$ is the proteome fraction of unnecessary genes and $\phi_{\max}$ is the total proteome fraction available for growth ($\phi_{\max} \approx 0.5$ for E. coli in exponential growth)[37,38]. A single gene with average expression level encodes a proteome fraction $\phi_U \sim 1/g$; this leads to a metabolic cost $\mathcal{L}'_{\text{phys}}(g) = (\lambda_0 - \lambda)/\lambda_0 \sim 1/(g\phi_{\max})$ per generation. Similarly, the energetic cost of a gene is of order $1/g$[53]. While the precise form of these cost components depends on details of cell metabolism, we expect generically $\mathcal{L}'_{\text{phys}}(g) \approx 1/g$. For evolution under phenotypic interference, this implies $\mathcal{L}'_{\text{phys}}(g) \lesssim \tilde{\sigma}$ for $g \gtrsim 5000$, which is similar to the interference load per gene in a standard microbe but becomes subleading in larger genomes.

The physiological cost per gene acts as a selective force on changes of genome size within a coalescence interval $\tilde{\sigma}^{-1}$. The inequality $\mathcal{L}'_{\text{phys}}(g) \lesssim \tilde{\sigma}$ says that such changes are weakly selected and suggests a two-scale evolution of genome sizes. On short time scales, the dynamics of gene numbers is permissive and allows the rapid acquisition of adaptive genes. On longer time scales (of order $\tau$; see Eq. 11 below), the interference load prunes marginally relevant genes in a more stringent way, for example, by invasion of strains with more compact genomes.

**Interference drives gene loss**. The near-neutral dynamics of genome size extends to gene losses, which become likely when a gene gets close to the inflection point of the sigmoid fitness landscape and the stability condition underlying Eq. 2 no longer holds (Fig. 4a). The relevant threshold gene selection, $f_0^c$, is

$$f_0^c \sim 2\tilde{\sigma} = \frac{4ug}{c} \qquad (10)$$

in the minimal model; see Eq. 5. Strongly selected genes ($f_0 \gg f_0^c \sim 2\tilde{\sigma}$) have equilibrium trait values firmly on the concave part of the landscape, resulting in small loss rates of order $u\exp(-f_0/2\tilde{\sigma})$[10]; these genes can be maintained over extended evolutionary periods. Marginally selected genes ($f_0 \lesssim f_0^c \sim 2\tilde{\sigma}$) have near-neutral loss rates of order $u$[10], generating a continuous turnover of genes. According to Eq. 10, the threshold $f_0^c$ for gene loss increases with genome size, which expresses again the evolutionary constraint on genome complexity. The dependence of the gene loss rate on $f_0$ and $\tilde{\sigma}$ is confirmed by simulations (Fig. 4b). The housekeeping coalescence rate $\tilde{\sigma} = 2ug/c$ sets a

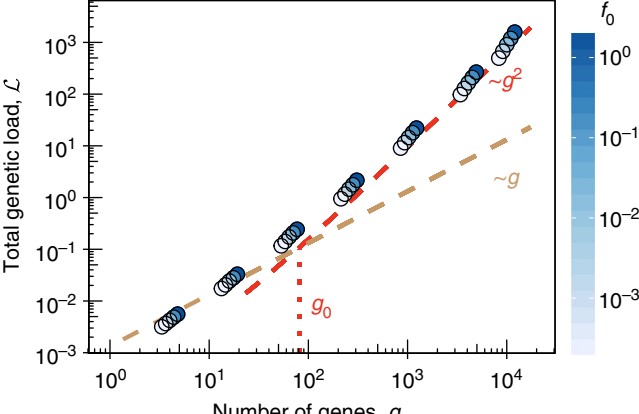

**Fig. 3** Genetic load. Total genetic load $\mathcal{L}$ versus the number of genes $g$ for asexual evolution for the minimal biophysical model at steady-state asexual evolution. Simulation results (circles) for different values of $f_0$ (indicated by color, as in Fig. 2); model results for independently evolving genes, $\mathcal{L} = ug$ (brown dashed line), and phenotypic interference load, $\mathcal{L}_{\text{int}} \sim ug^2/c$ (red dashed line); see Eq. 9. The superlinear behavior of $\mathcal{L}$ for $g > g_0 \sim 10^2$ indicates strong selection against genome complexity. Other simulation parameters as in Fig. 2; see Methods section for simulation details

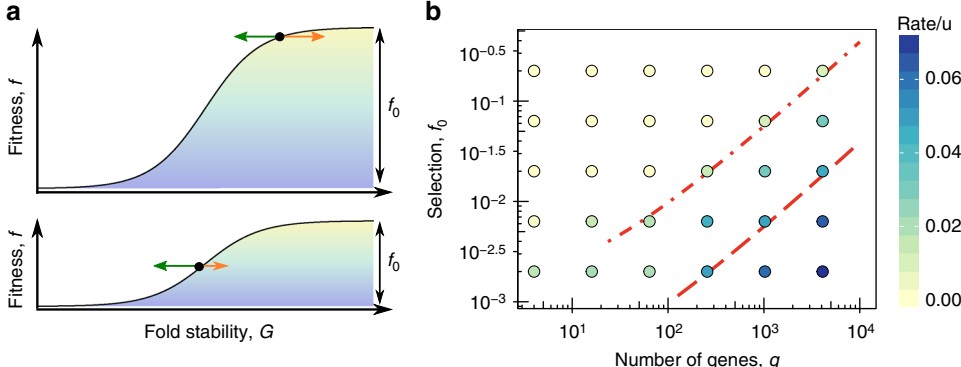

**Fig. 4** Gene loss. **a** The stability-dependent fitness landscape $f(G)$ of Fig. 1a is shown for two values of the selection amplitude $f_0$, which is the fitness difference between functional and dysfunctional proteins. Arrows indicate the effect of deleterious mutations (green) and selection (orange, proportional to $f_0$) on the population mean trait. For strongly selected genes ($f_0 \gg \tilde{\sigma}$, top panel), the mutation-selection dynamics generates stable equilibria; cf. Eqs. 2, 23. For weakly selected genes ($f_0 \lesssim \tilde{\sigma}$, lower panel), mutations outweigh selection, leading to significant gene loss. **b** Rate of gene loss (indicated by color, in units of $u$) as a function of the gene selection coefficient, $f_0$, and the number of co-evolving genes $g$. Genes with $f_0 \sim \tilde{\sigma}$ (long-dashed line, cf. Fig. 2b) have appreciable loss rates; genes with $f_0 \gtrsim 10\tilde{\sigma}$ (dashed-dotted line) have negligible loss rates, i.e., are conserved under phenotypic interference. Other simulation parameters as in Fig. 2; see Methods section for simulation details

lower bound for $f_0^c$, adaptive evolution can lead to much larger values of $\tilde{\sigma}$ and $f_0^c$.

**Load accumulation in evolution experiments**. After a change in gene number or other systems parameters, the evolutionary process reaches a new steady state. Because (additional) deleterious trait changes are only marginally selected (i.e., have selection coefficients of magnitude $|s| \lesssim \tilde{\sigma}$), the relaxation time $\tau$ is of the order of the inverse mutation rate per trait,

$$\tau \sim \frac{1}{u} = \frac{2g}{c}\frac{1}{\tilde{\sigma}}, \qquad (11)$$

where we have used Eq. 5. This time scale exceeds the coalescence time $\tilde{\sigma}^{-1}$ and is of order $10^6$ generations for a standard microbe. Hence, interference selection against complexity is a potent evolutionary force affecting natural populations but is beyond the time scales of laboratory evolution experiments.

Nevertheless, the phenotypic interference model makes testable predictions on load accumulation in laboratory populations. Consider a standard microbe that has an initial housekeeping interference load $\mathcal{L}_{int}/g = 2\tilde{\sigma}$ per gene and is subject to strong adaptive pressure in the experiment, generating an increased coalescence rate $\tilde{\sigma}_{ad} \gg \tilde{\sigma}$. Equations 6, 11 then predict a lower bound for the genome-wide rate of load increase, $\mathcal{L}_{int} \gtrsim 2ug\tilde{\sigma}$ per generation. This loss reflects the system-wide collateral degradation of protein stability, which is caused by deleterious hitchhiker mutations of the adaptive process.

A collateral fitness decline of this type and magnitude has been observed in *E. coli* populations from long-term evolution experiments[20–23]. While the decline is masked in the original long-term experiments by a larger adaptive fitness gain[21], it has been revealed by fitness measurements of the evolved strains on other substrates[20]. A substantial part of the fitness loss can be rescued in fitness assays at lower temperature, suggesting a link to protein stability[20]. The phenotypic interference model supports this interpretation. Protein stability $G$, as well as quantitative protein function traits, provides a large, genome-wide supply of weakly selected mutations prone to hitchhiking ($s \lesssim \tilde{\sigma}$). Moreover, the biophysical fitness landscapes of protein stability and affinity are explicitly temperature-dependent, which explains why fitness losses by deleterious mutations can be compensated by temperature reduction. We obtain a lower bound on the fitness loss related to the genome-wide attrition of these biophysical

traits, $\dot{\mathcal{L}} \sim 10^{-5}$ per generation, by evaluating the temperature-rescuable part of the fitness decline in mutator lines (see Methods section). Nonsynonymous substitutions have been observed at a genome-wide rate $ug \sim 10^{-2}$ per generation in these lines, and a large part appears to be effectively neutral hitchhikers[22]. Associating these substitutions with quantitative traits, the phenotypic interference model provides a lower-bound estimate $\dot{\mathcal{L}}_{int} \sim 2 \times 10^{-6}$ per generation (see Methods section), which is consistent with the observed loss rate.

**The pathway to sexual evolution**. Recombination reshuffles genome segments at a rate $R$ per genome and per generation ($R$ is also called the genetic map length). Evolutionary models show that recombination generates linkage blocks that are units of selection. A block contains an average number $\xi$ of genes, such that there is one recombination event per block and per coalescence time, as given by the relation $R\xi/(g\tilde{\sigma}(\xi)) = 1$[13,15,54,55]. Depending on $R$, these models predict a regime of asexual evolution, where selection acts on entire genotypes ($\xi \sim g$), and a distinct regime of sexual evolution with selection acting on individual alleles ($\xi \ll g$). Here we focus on the evolution of the recombination rate itself and establish a selective avenue for the transition from asexual to sexual evolution.

With the phenotypic interference scaling $\tilde{\sigma}(\xi) = 2u\xi/c$ for $\xi \gtrsim c$, as given by Eq. 5, our minimal model produces an instability at a threshold recombination rate

$$R^* = \tilde{\sigma} = \frac{2ug}{c}, \qquad (12)$$

signaling a first-order phase transition with the genetic load as order parameter. For $R < R^*$, the population is in the asexual mode of evolution ($\xi \sim g$), where phenotypic interference produces a superlinear load $\mathcal{L}_{int} = 2ug^2/c$. For $R > R^*$, efficient sexual evolution generates much smaller block sizes ($\xi \sim c$). In this regime, the load drops to the linear form $\mathcal{L}_0 = ug \ll \mathcal{L}_{int}$ providing a net long-term evolutionary fitness gain $\Delta\mathcal{L} = \mathcal{L}_{int} - \mathcal{L}_0 \simeq \mathcal{L}_{int}$. The first-order transition is a specific consequence of phenotypic interference. Because recombination rate and coalescence rate in a linkage block are both proportional to the block size $\xi$, the recombination-coalescence balance criterion takes the $\xi$-independent form $R^*/g = 2u/c$. That is, linkage blocks cover either the entire genome ($\xi \sim g$) or just small genome segments ($\xi \sim c$). The resulting drop of $\mathcal{L}$ in

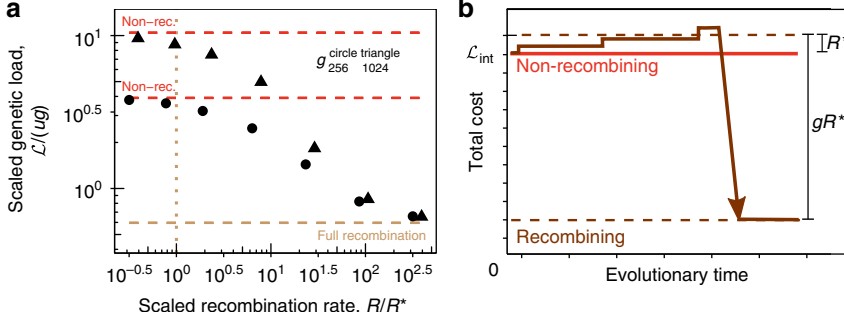

**Fig. 5** Transition to sexual evolution. **a** Scaled genetic load, $\mathcal{L}/(ug)$, versus scaled recombination rate, $R/R^\star$, for different genome sizes. The observed load rapidly drops from the superlinear scaling of phenotypic interference, $\mathcal{L}_{int} = 4ug^2/c$ (asymptotic data: red lines), to the linear scaling of unlinked genes, $\mathcal{L}_0 \sim ug$ (brown line). This signals a transition to sexual evolution around the threshold recombination rate $R^\star = 2ug/c$ (dotted line); the transition is rounded by finite size effects in population size and genome size. Other simulation parameters as in Fig. 2; see Methods section for simulation details and Supplementary Table 2 for a comparison with biological data. **b** Selective pathway for the evolution of sex (schematic). Starting from a non-recombining state with interference load $\mathcal{L}_{int}$ (red line), a lineage evolving a recombination rate $R \sim R^\star$ (brown line) incurs a weak cost $\mathcal{L}_{rec} \sim R^\star$. Subsequently, this lineage can eliminate its interference load, leading to a large benefit $\Delta\mathcal{L} \sim gR^\star$, and outcompete non-recombining lineages in a common ecological niche

recombining populations close to $R^\star$ is confirmed by simulations (Fig. 5a). The process of recombination comes with a direct, short-term cost $\mathcal{L}_{rec}$ per generation, which includes mating costs, physiological costs, and deleterious reshuffling costs, and can potentially prevent the evolution of recombination. The classical factor 2 scenario of obligately sexual populations says that this cost is of order 1 per recombination event; that is, $\mathcal{L}_{rec} = R$[28,56,57]. For early isogamous populations without the full machinery of sexual reproduction, $\mathcal{L}$ is likely to be smaller[58]. Importantly, in the phenotypic interference mode, this cost remains always marginal. Even the upper-bound assumption $\mathcal{L}_{rec} = R$ leads to a cost $\mathcal{L}_{rec} \lesssim R^\star = \tilde{\sigma}$ at the transition, which implies only weak negative selection.

Together, the theory of phenotypic interference suggests a specific selective pathway for the evolution of recombination (Fig. 5b). First, given that the evolution of recombination at a rate of order $R^\star$ is near-neutral, a recombining sub-lineage with $R \sim R^\star$ arising in an asexual background population can fix by genetic drift and draft. Second, a recombining strain with $R > R^\star$ can eliminate the interference load by the parallel fixation of beneficial mutations in unlinked genome segments. This leads to a long-term benefit $\Delta\mathcal{L} \sim gR^\star = g\tilde{\sigma}$ over non-recombining but otherwise equivalent strains; by Eq. 9, this benefit is of order 1 for a standard microbe. Hence, the evolved recombining strain can readily outcompete related non-recombining strains in the same ecological niche. The threshold recombination rate $R^\star$ is of the order of the genome-wide mutation rate $ug$, so even rare facultative recombination provides a robust pathway to sexual evolution. This pathway builds on a separation of selection scales: the near-neutral establishment of recombination is followed by the buildup of a large benefit. We can compare observed recombination rates in natural populations with the predicted threshold rates $R^\star$ (Supplementary Table 2). Consistently, genome-wide average rates for species in different parts of the tree of life are always well above $R^\star$; a high-resolution recombination map of the *Drosophila* genome shows low-recombining regions with values above but of order $R^\star$[59,60].

The phenotypic interference pathway to recombination has highly universal characteristics: its long-term benefit of recombination is $g$-fold higher than the upper-bound cost, independently of details of the genome-wide selection and mutation landscape. In particular, this pathway does not require any of the strong assumptions of previous models for the evolution of recombination, which include direct benefits of recombination[28,58,61,62],

strong and continual adaptation[61,63–65], and genome-wide epistasis between mutations[28,65–68]. It builds instead on local diminishing-return epistasis for functional traits of individual proteins, which is a natural consequence of their underlying biophysical mechanism. Recent fitness-wave models, which have an interference dynamics qualitatively similar to ours, quantify the difference in adaptive speed between clonally evolving and recombining populations[7–9,11–13], but a direct cost-benefit balance of recombination based on genetic load has not been attempted. We note that these models assume mutations with a fixed distribution of selection coefficients and no local epistasis, which creates important quantitative differences to phenotypic interference. First, strongly deleterious effects of asexual evolution, which are associated with the onset of Muller's ratchet, set in at larger genome sizes[8] than under phenotypic interference (Fig. 1c). Second, the crossover to sexual evolution, which has been studied in the context of adaptive fitness waves, takes place at a larger recombination rate $R$[15,69] and, hence, a larger recombination cost. A more detailed model comparison is given in Supplementary Discussion.

## Discussion
Here we have developed the evolutionary genetics of multiple biophysical traits in non-recombining populations. Our approach combines quantitative trait theory with fitness wave theory. We find a specific evolutionary mode of phenotypic interference, which is characterized by a feedback between global and local selection. The system-wide genetic variation of the traits generates fitness variance, which, in turn, determines the scale of selection at local genomic sites encoding the traits. This feedback generates highly universal features, which do not depend on system details. These include the complexity of the evolutionary process and the scaling of coalescence rate and genetic load with the gene number, as given by Eqs. 5–9. A similar destructive feedback generating a superlinear cost has been identified in crosstalk of gene regulation[70]. Importantly, phenotypic interference also generates universal local selection. By Eq. 3, the average selective amplitude of trait-changing mutations decouples from the total fitness effect of the trait. That is, the spectrum of site selection coefficients is not a fixed input, but a dynamical output of the evolutionary process. This selection filter is the main difference of our approach to previous population-genetic models of asexual evolution[5–13]. We argue it is a relevant step towards biological realism.

Phenotypic interference depends on two prerequisites: selection is globally clonal and its local genomic units are broadly epistatic. The clonality of selection is a generic consequence of low recombination rates; broad fitness epistasis is a ubiquitous feature of biophysical gene traits, including protein stabilities and activities. Such traits have non-linear fitness landscapes, in which the selection on trait changes depends on the trait value (Fig. 1a).

We have shown that phenotypic interference produces systems-biological effects on different evolutionary time scales. In clonal adaptation experiments, it predicts a system-wide functional and fitness degradation in line with observations[20–23]. On macro-evolutionary scales, it generates strong selection against genome complexity in clonally reproducing populations. The underlying genetic load originates from the interference of phenotypic variants within a population and accumulates with a time delay beyond the coalescence time, as given by Eq. 11. Interference load acts as an evolutionary force in an ecological context: microbial strains with shorter genomes can outcompete otherwise similar strains with longer genomes that are in the same ecological niche. We have shown that this force, which arises naturally from a systems perspective of multiple biophysical traits, provides a robust eco-evolutionary pathway for the transition to recombination. Its selective input is local fitness epistasis, which occurs ubiquitously in quantitative molecular traits. Therefore, unlike previous models based on global epistasis[28,65–68], this pathway does not require ad-hoc assumptions on the form of selection.

The target of phenotypic interference is molecular complexity, which can be regarded as a key systems-biological observable. In our simple biophysical models, we measure complexity by number of stability and affinity traits in a proteome. This is clearly just a starting point towards a broader systems-biological approach that includes regulatory, signaling, and metabolic networks. These define additional landscapes of biophysical interactions, but the key evolutionary mechanisms of phenotypic interference—globally clonal selection and tuned, epistatic selection on system components—are expected to play out in a similar way. In a systems model, we can define complexity as the number of (approximately) independent molecular quantitative traits, which includes network contributions that scale in a nonlinear way with genome size. Interference selection affects the complexity and architecture of all of these networks, establishing new links between evolutionary and systems biology to be explored in future work.

## Methods

**Biophysical fitness models.** In thermodynamic equilibrium at temperature $T$, a protein is folded with probability $p_+(G) = 1/[1 + \exp(-G/k_B T)]$, where $G$ is the Gibbs free energy difference between the unfolded and the folded state and $k_B$ is Boltzmann's constant. A minimal biophysical fitness model for proteins takes the form

$$f(G) = f_0 p_+(G) + C = \frac{f_0}{1 + \exp(-G/k_B T)} + C \quad (13)$$

with a single selection coefficient capturing functional benefits of folded proteins and metabolic costs of misfolding[32–34]. The constant $C$ is irrelevant for the computation of fitness differences (selection coefficients). This model describes the effect of a protein on Malthusian (logarithmic) fitness, depending on its free energy of folding. Similar fitness models based on binding affinity have been derived for transcriptional regulation[29,30,71,72]; the rationale of biophysical fitness models has been reviewed in refs. [36,73]. Equation 13 applies to genes with individually small fitness effects ($f_0 \ll 1$). An appropriate extension to essential genes is a landscape describing zero growth (lethality) at a finite stability threshold $G_0$, which corresponds to a singularity of the Malthusian fitness, $f(G) \rightarrow -\infty$ for $G \rightarrow G_0$. An example is the landscape $f(G) = \log[f_0/(1 + \exp(-G/k_B T)) + (1 - f_0)]$, which has a threshold $G_0$ given by $p_+(G_0) = 1 - 1/f_0$ for $f_0 > 1$; alternative models for essential genes are described in refs. [31,32]. However, the extended fitness landscape retains the form Eq. 13 in the regime of stable folding ($G/k_B T \gtrsim 1$), which implies that our conclusions remain unaffected. In particular, the load per gene remains independent of the selection amplitude $f_0$, as given by Eq. 9 and confirmed by simulations (Fig. 3). In Supplementary Methods 3, we introduce further alternative

fitness landscapes for proteins and show that our results depend only on broad characteristics of these landscapes.

The minimal global fitness landscape for a system of $g$ genes with traits $G_1, \ldots, G_g$ and selection coefficients $f_{0,1}, \ldots, f_{0,g}$ is taken to be additive, i.e., without epistasis between genes,

$$f(G_1, \ldots, G_g) = \sum_{i=1}^{g} \frac{f_{0,i}}{[1 + \exp(-G_i/k_B T)]}. \quad (14)$$

**Evolutionary model.** We characterize the population genetics of an individual trait $G$ by its population mean $\Gamma$ and its expected variance $\Delta_G$. These follow the stochastic evolution Equations[45]

$$\dot{\Gamma} = -u\kappa\epsilon_G + \Delta_G f'(\Gamma) + \chi_\Gamma(t), \quad (15)$$

$$\dot{\Delta}_G = \left(u\epsilon_0^2 - N_e^{-1}\Delta_G\right)\Delta_G + \Delta_G^2 f''(\Gamma) + \chi_\Delta(t). \quad (16)$$

These equations contain white noise $\chi_\Gamma(t)$ of mean $\langle\chi_\Gamma(t)\rangle = 0$ and variance $\langle\chi_\Gamma(t)\chi_\Gamma(t')\rangle = (\Delta_G/2N_e)\delta(t - t')$ and $\chi_\Delta(t)$ of mean $\langle\chi_\Delta(t)\rangle = 0$ and variance $\langle\chi_\Delta(t)\chi_\Delta(t')\rangle = (2\Delta_G^2/2N_e)\delta(t - t')$ with an effective population size $N_e = 1/2\tilde{\sigma}$ generated by genetic draft. This dynamics is characterized by the rate $u$, the mean effect $(-\kappa)\epsilon_G$, and the mean square effect $\epsilon_G^2$ of trait-changing mutations. We use effects $\epsilon_G \approx 1 - 3k_B T$, which have been measured for fold stability[31,74] and for molecular binding traits[29,75,76]. Furthermore, we approximate the mutational bias $\kappa(\Gamma)$ by a constant $\kappa = 1$, which reflects the observation that most mutations affecting a functional trait are deleterious.

**Evolutionary equilibria for individual traits.** We now derive the equilibrium conditions of the model given by Eqs. 15, 16, which are used in the main text. This involves three steps. First, the deterministic term in Eq. 16 determines the average trait diversity $\Delta_G$ as given in Eq. 1, if we neglect the selection component (this will be justified in step three below). That is, $\Delta_G$ follows from a mutation-coalescence balance: the trait gains a heritable variance $\Delta_G$ by new mutations at a speed $u\epsilon_G^2$, and it loses variation by coalescence at a rate $2\tilde{\sigma}$. Equation 1 is consistent with well-known results for the average sequence diversity $\Delta$, indicating that diversity expectation values do not depend on details of the coalescence process. These results include the relation $\Delta = 4uN_e$ in the standard theory of neutral evolution, where $N_e$ is proportional to the actual population size[40]. The same relation is obtained for the sequence diversity of neutral genomic sites in models of genetic draft[41] and in fitness wave models, where $N_e = (2\tilde{\sigma})^{-1}$ is determined by selection[14,42]. To obtain the equivalent form for a quantitative trait $G$, we simply rescale the sequence diversity by the mean square effect $\epsilon_G^2$[44,45], which leads to Eq. 1.

Second, the equilibrium point of the mean trait $\Gamma$ follows from a mutation-selection balance, as given by Eq. 2. The rate of stability increase by selection, $(\partial\Gamma/\partial t)_{sel.} = \Delta_G f'(\Gamma)$, is essentially a statement of Fisher's theorem; the corresponding rate of fitness increase reads

$$\left(\frac{\partial f}{\partial t}\right)_{sel.} = f'(\Gamma)\left(\frac{\partial\Gamma}{\partial t}\right)_{sel.} = \Delta_G f'^2(\Gamma) = \Delta_f(\Gamma). \quad (17)$$

The rate of stability decrease by mutations is the product of the total mutation rate per trait, $u$, and the mean effect per mutation $(-\kappa)\epsilon_G$ with the approximation $\kappa = 1$ as discussed above. In Supplementary Methods 1 and Supplementary Fig. 3, we derive the equilibrium of the mean trait $\Gamma$ in a fully stochastic calculus. We also note that the weakness of stabilizing selection on the trait diversity is consistent with finite directional selection on the population mean trait[45].

Third, we can check a posteriori that the selection term in Eq. 16 can be self-consistently neglected. For stable genes, our biophysical traits live on the downward-curved shoulder of the fitness landscape (where $f''(G) < 0$). The neutral relation (1) remains approximately valid for these traits if the resulting stabilizing selection on the trait diversity is negligible. This condition can be written in terms of the diversity load $\mathcal{L}_\Delta \equiv f(\Gamma) - \bar{f}$,

$$\frac{\mathcal{L}_\Delta}{\tilde{\sigma}} \simeq \frac{\Delta_G|f''(\Gamma)|}{\tilde{\sigma}} \lesssim 1; \quad (18)$$

see ref. [45]. We now show that this condition is self-consistently fulfilled throughout the phenotypic interference regime. Evaluating the expected fitness curvature in the high-fitness part of the minimal fitness landscape, Eq. 13, where $f''(\Gamma) = -f'(\Gamma)/k_B T$, and in the mutation-coalescence equilibrium given by Eq. 1, we obtain $f'' = -2\tilde{\sigma}/(\epsilon_g k_B T)$. By Eqs. 6, 18 then reduces to

$$\frac{\Delta_G}{\epsilon_G^2} = \frac{c}{4g} \lesssim 1, \quad (19)$$

which is identical to the condition for phenotypic interference, Eq. 8. We conclude that Eq. 1 is a valid approximation for the trait diversity throughout the phenotypic interference regime. This is confirmed by our simulation results (Supplementary Fig. 2a).

**Housekeeping equilibrium and fitness waves of phenotypic interference**. The deterministic equilibrium solution ($\dot{\Gamma} = 0$, $\chi = 0$) of Eq. 15 determines the dependence of $\Delta_G$ and the associated fitness variance $\Delta_f = \Delta_G f'^2(G)$ on $\tilde{\sigma}$, as given by Eq. 3; the same scaling follows from the full stochastic equation (Supplementary Methods 1). The derivation of the global housekeeping steady state, Eqs. 5–7, uses two additional inputs: the additivity of the fitness variance, $\sigma^2 = g\Delta_f$, which is confirmed by our simulations (Supplementary Fig. 4), and the universal relation Eq. 4 in a fitness wave[12,13]. This relation is obtained by evaluating the total fitness span $\hat{\sigma} \equiv f_{max} - f_0$ in a population of finite census size $N$. Here $f_{max}$ is the fitness maximum in the set of established mutations (i.e., mutations that have overcome genetic drift), which requires a mutant clone frequency $x \gtrsim 1/(N(f - f_0))$. Given a Gaussian bulk fitness distribution $\rho(f) = (2\pi\sigma^2)^{-1/2} \exp[-(f - f_0)^2/2\sigma^2]$, the tail condition for established mutations, $\int_{f_{max}}^{\infty} \rho(f)\, df \sim 1/(N\hat{\sigma})$, produces $\hat{\sigma}^2/\sigma^2 \sim \log(N\sigma)$. Equation 4 then follows via the kinematic relation $\tilde{\sigma} = \sigma^2/\hat{\sigma}$ given by Fisher's theorem. The prefactor $c_0$ is model-dependent and known only in the infinitesimal fitness wave limit, e.g., $c_0 \sim 100$ in the model of refs. [12,13]. Here we treat $c_0$ as a fit parameter in simulations. The wave parameter $c$ has a double interpretation in generic fitness wave models: it relates the total fitness span and the coalescence time to the fitness variance, $\hat{\sigma}^2 = c\sigma^2$ and $N_e^2 = \tilde{\sigma}^{-2} = c\sigma^{-2}$. The dependence of $c$ on genome size under phenotypic interference, Eq. 7, is obtained by inserting Eqs. 5 into 4 and neglecting subleading terms $\mathcal{O}(\log\log(Nug))$. It is important to note that the housekeeping fitness wave describes a genome-wide mutation-selection steady state of constant mean fitness and without adaptive changes[12,77], which is consistent with the equilibria of deleterious and beneficial substitutions in each gene[30].

**Local and global diversity scaling under phenotypic interference**. Equation 19 expresses an important scaling property of the phenotypic interference regime: individual traits evolve in the low-mutation regime and are monomorphic at most times. In contrast, the cumulative variance of all traits defines a polymorphic fitness wave,

$$\frac{4g\Delta_G}{\epsilon_G^2} = c \gtrsim g_0, \qquad (20)$$

where we used Eq. 19. A related measure is the complexity of the fitness wave, defined as the average number of beneficial substitutions per coalescence time, $g\langle v_+\rangle/\tilde{\sigma} = (g/\tilde{\sigma})\int_0^{\infty} \nu(s) v_+(s)\, ds$. Here $\nu(s)$ is the spectrum of site selection coefficients, which has the average $2\tilde{\sigma}$ by Eq. 3, and $v_+(s)$ is the equilibrium beneficial substitution rate at a site of selection coefficient $s$, which has a near-neutral regime $v_+(s) \simeq u/2$ for $s \lesssim \tilde{\sigma}$ and rapidly decreases for $s \gtrsim \tilde{\sigma}$. Hence, we obtain a wave complexity

$$\frac{g\langle v_+\rangle}{\tilde{\sigma}} \approx \frac{ug}{2\tilde{\sigma}} = \frac{c}{4} \qquad (21)$$

with a prefactor of order 1; here we have used Eq. 5. By Eq. 7, the fitness wave measures Eqs. 20, 21 depend only weakly on $g$.

**Onset of phenotypic interference**. Interference effects on quantitative traits can be read off from the scaling of the genetic load, which has the linear form $\mathcal{L} = ug$ for independently evolving genes and is given by Eq. 9 in the phenotypic interference regime. Equating these relations identifies an onset gene number $g_0$ given by

$$ug_0 = \frac{4ug_0^2}{c}, \qquad (22)$$

or equivalently by Eq. 8.

**Evolutionary equilibria of stable genes**. Equilibrium traits of genes with $f_0 \gg \tilde{\sigma}$ are located in the high-fitness part of the minimal fitness landscape, $f \simeq f_0[1 - \exp(-G/k_B T)]$. These genes have an average fitness slope

$$f' = \left(\frac{\Delta_f}{\Delta_G}\right)^{1/2} = \frac{2\tilde{\sigma}}{\epsilon_G}, \qquad (23)$$

an average trait $\Gamma = k_B T \log(f_0\epsilon_G/2\tilde{\sigma} k_B T) > 0$, and an average load $\mathcal{L}_{int}(g)$ given by Eq. 9. This is in accordance with well-known population data of protein stability in microbial populations[34]: typical genes balance a few $k_B T$ above the melting point $G = 0$, which corresponds to the shoulder of the fitness landscape above the inflection point (Fig. 1a). The average stability has only a log-dependence on evolutionary rates.

**Phenotypic interference in adaptive evolution**. Here we show that the phenotypic interference scaling extends to simple models of adaptive evolution. In the minimal biophysical model, we assume that protein stabilities are still at local evolutionary equilibria of the universal form given by Eq. 3, generating a combined housekeeping component of the fitness variance, $\sigma_{hk}^2 = g\Delta_f = 2gu\tilde{\sigma}$. The global fitness variance acquires an additional contribution from adaptive evolution of

other system functions,

$$\sigma^2 = c\tilde{\sigma}^2 = \sigma_{hk}^2 + \phi, \qquad (24)$$

where $\phi$ is the adaptive fitness flux or rate of adaptive fitness gain[78]. This term quantifies the deviations of the adaptive evolutionary process from housekeeping evolution. Closure of the modified dynamics leads to an increased coalescence rate

$$\tilde{\sigma} = \frac{ug}{c}\left(1 + \sqrt{1 + \frac{c\phi}{g^2 u^2}}\right) \qquad (25)$$

and total interference load

$$\mathcal{L}_{int} = 2g\tilde{\sigma} = \frac{2ug^2}{c}\left(1 + \sqrt{1 + \frac{c\phi}{g^2 u^2}}\right). \qquad (26)$$

Hence, the load retains the leading nonlinearity generated by housekeeping evolution, as given by Eq. 9; this is true even if we assume that $\phi$ is proportional to $g$. At high fitness flux ($\phi \gtrsim g^2 u^2/c$), coalescence becomes dominated by adaptation, leading to a further substantial decrease in the efficacy of selection. This is the likely regime of the laboratory evolution experiments discussed in the main text.

**Fitness loss in evolution experiments**. Bacterial lineages from the long-term evolution experiment of ref. [21] have been subject to fitness measurements in diverse environments[20]. These measurements show heterogeneous combinations of environment-specific fitness gains and losses compared to the ancestor strain. In mutator lines evolved over 50,000 generations a higher average growth rate $\lambda$ at temperature 30 °C than at temperature 37 °C. To extract a bona fide order-of magnitude estimate of the fitness loss due to attrition of quantitative traits, we evaluate the population-average difference in log growth rate, $\Delta L = \langle \log(\lambda_{30°}/\lambda_{37°})\rangle = 0.47/50\,k$ generations, using the data provided in ref. [79]. The observed average number of fixations per stable population clade is about 500/50 k generations[22]. These data provide the estimates $\dot{\mathcal{L}} \approx 10^{-5}$ and $ug \approx 10^{-2}$ used in the main text, and they inform the model estimate $\dot{\mathcal{L}}_{int} \sim 2ug\tilde{\sigma}$ with the standard microbe housekeeping value $\tilde{\sigma} \sim 10^{-4}$. We note two additional consistency checks: (a) The inferred average deleterious fitness effect per substitution, $s = \dot{\mathcal{L}}/(ug) \approx 10^{-3}$ is of order of the observed inverse coalescence time, supporting the conclusion that a large fraction of these changes is effectively neutral[22]. (b) Non-mutator lines, which have a 100-fold lower mutation rate, do not show evidence of a large proportion of effectively neutral fixations and have significantly lower $\Delta L$.

**Numerical simulations of phenotypic interference**. We use a Wright-Fisher process to simulate the evolution of stability traits in a population. A population consists of $N$ individuals with genomes $\mathbf{a}^{(1)}, \ldots, \mathbf{a}^{(N)}$. A genotype $\mathbf{a} = (\mathbf{a}_1, \ldots, \mathbf{a}_g)$ consists of $g$ segments; each segment is a subsequence $\mathbf{a}_i = (a_{i,1}, \ldots, a_{i,\ell})$ with binary alleles $a_{j,k} = 0, 1$ ($i = 1, \ldots, g; k = 1, \ldots, \ell$). A segment $\mathbf{a}$ defines a stability trait $G(\mathbf{a}) = \sum_{k=1}^{\ell} \mathcal{E}_k a_k + G_0$, where $G_0$ is the minimum trait value. The resulting effect distribution of point mutations has as a second moment $\epsilon_G^2 = \sum_{k=1}^{\ell} \mathcal{E}_k^2/\ell$ and a first moment $\kappa_0\epsilon_G = \sum_{k=1}^{\ell} \mathcal{E}_k(1 - 2\langle a_k\rangle)/\ell$, where $\langle a_k\rangle$ is the state-dependent probability of a mutation at site $k$ being beneficial and brackets $\langle . \rangle$ denote averaging across parallel simulations or time. The genomic fitness is $f(\mathbf{a}) = \sum_{i=1}^{g} f(G(\mathbf{a}_i); f_{0,i})$ with $f(G)$ given by Eq. 13 and gene-specific amplitudes $f_{0,i}$. In each generation, the sequences undergo point mutations with probability $\mu\tau_0$ for each site, where $\tau_0$ is the generation time, and the sequences of the next generation are drawn by multinomial sampling with a probabilities proportional to $1 + \tau_0 f(\mathbf{a})$.

Simulations are performed with parameters $N = 1000$, $N\mu = 0.0125$, each trait with genomic base of size $\ell = 100$, and each site with equal effect $E_k = 1$. The population size $N$ is smaller than in natural populations; this is compensated by an increased mutation rate to keep the product $N\mu$ at a realistic value. The quantitative trait dynamics is insensitive to the form of the effect distribution[45,80]. To increase the performance of the simulations, we do not keep track of the full genome. We only store the number of deleterious alleles $n_i = \sum_{k=1}^{\ell} a_{i,k}$ for each trait, we draw mutations with rate $u = \mu\ell$, and we assign to each mutation a beneficial change $\mathcal{E}$ with probability $n_i/\ell$ and a deleterious change $-\mathcal{E}$ otherwise. This procedure produces the correct genome statistics for bi-allelic sites with uniform trait effects $\mathcal{E}_i = \mathcal{E}$. Simulation data are shown with theory curves for $\kappa = 1$, which provide a good fit to all amplitudes; the input $\kappa_0$ is different by a factor of order 1 which includes fluctuation effects (Supplementary Methods 1).

Simulations run to reach a stationary state and then have 2000–128,000 consecutive measurements (for largest $g = 4096$ to smallest $g = 4$) every 400 generations. These intervals exceed the correlation time of the coalescence process. Therefore, measurements of the global observables $\sigma^2$, $\tilde{\sigma}$, and $\mathcal{L}$, as well as the local variance $\delta_g$ decorrelate. Measurements of the other local variables $s^2$, the loss rate, and $\Delta_f$ are averaged over all $g$ genes.

For the simulations of housekeeping evolution in Figs. 2, 3, where we are not explicitly interested in the loss of genes, we use an exponential approximation of the stable regime of the stability fitness landscape. The reason is a limited accessible parameter range in simulations constraining the values of $f_0$ and $\tilde{\sigma}$ due to finite $N$. We checked that the exponential approximation gives the same results as the full

model in the regime $f_0/\tilde{\sigma} \gg 1$, where the gene loss rate in the biophysical landscape is negligible.

For the loss rate measurements of Fig. 4b, a long-term stationary population is maintained by evolving 70% of the traits in a biophysical fitness landscape with selection $f_0$; the remaining 30% of the traits are modeled to be essential with selection $10f_0$. Gene loss is defined by the condition $G < -3.5k_BT$. To maintain a constant number of genes, lost genes are replaced immediately with an input trait value $G > 0$.

For simulations with recombination (Fig. 5a), we draw recombination events with rate $NR$ for the whole population from a Poisson distribution. Each recombination event is implemented as one crossover between the genomes of two individuals at a random, uniformly distributed position of the genomes.

**Reporting summary**. Further information on research design is available in the Nature Research Reporting Summary linked to this article.

## Data availability
The data generated from the simulations are available from the corresponding author upon reasonable request.

## Code availability
The code for the simulations of this study is included as Supplementary Software 1.

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

## Acknowledgements
We thank T. Bollenbach and A. Sousa for discussions. This work has been supported by Deutsche Forschungsgemeinschaft grants SFB 680 and SFB 1310 (to M.L.). We acknowledge computational support by the CHEOPS platform at University of Cologne.

## Author contributions
Conceptualization, all; Methodology, all; Software, T.H. and D.K.; Validation, all; Formal analysis, all; Investigation, all; Writing, all; Visualization, all; Supervision, M.L.; Funding Acquisition, M.L.

## Additional information

**Competing interests:** The authors declare no competing interests.

