## [Peer Review File · Nature Communications]

Reviewers' comments:

Reviewer #1 (Remarks to the Author):

In this paper, the authors build a theory of "phenotypic interference". In short, this is a theory for the evolution of clonal populations in which (even housekeeping) selection for many additive traits (e.g. thermodynamic stability of proteins) results in a load that scales quadratically with the number of traits due to clonal interference. This mode of evolution has been previously unobserved, because (i) of neglecting biophysical realism due to nonlinear behavior of individual traits; (ii) of nontrivial coupling between the local (single trait) and global (fitness wave) levels of evolution.

I believe that this paper makes a fundamental contribution to the field and recommend that it is published.

My major recommendation is that the clarity of initial exposition, as I describe below, could be increased to make the paper parseable more widely than to the specialized audience dealing with clonal interference. Several concrete suggestions:

1) For someone not familiar with the concept of "fitness wave" in detail it is hard to parse the intro para "Over a wide range of model parameters..." What I would find more important is to highlight first the feedback between local and global selection and explain what this is in words. Here, or perhaps later (in "theory of phenotypic interference" initial para) it would really help to highlight how such feedback emerges: through the fact that the $\tilde{\sigma}$ (related to eff. pop size) is set by the evolutionary dynamics itself. This is interesting to wider field as by default eff. pop size is often just assumed to be some constant, which would be wrong in this context. This is, if I understand properly, the gist of point (ii) above;

2) I would also highlight more point (i) above, namely the necessity of treating the essential biophysical nonlinearity (figure 1); right now, this is said in caption to Fig 1 (where you compare biophys model in red and discrete fitness model in blue), but I suggest you highlight this when you call out Figure 1B.

3) As above, for "Theory of phenotypic interference" first para, prepare the reader how the local and global fitness dynamics will be coupled, i.e., by mentioning specifically that the local part will assume some $\tilde{\sigma}$, which will be then set by the global fitness wave part. This I understand only after reading through the whole manuscript first and being confused.

4) Minor, in the subsection "Evolution of a quantitative trait under interference selection" -- that subsection is just a general section on the evolution of a quantitative trait, as the interference doesn't come in yet, correct? Perhaps you can amend the title. Also: can you make clear what parts of this subsection are novel (e.g., relations in Eq (1) and (2) appear straightforward, has this been done before)?

5) I had the most trouble parsing the crucial section "Housekeeping evolution of multiple traits". First, it would be good to be more clear what are the necessary conditions for the fitness wave regime to hold in this case. Make it more explicit that the fitness wave theory for your case contributes the crucial equation that relates the fitness variance and the coalescence rate, and perhaps give a few sentence intuition behind it. Moreover, currently you assert $c_0 \sim 100$, which requires at least a few words about where the magic constant emerges from. Second, you refer to c as the "slowly varying parameter". In what sense is this slow, can you be more precise? Third, can you comment briefly about how Eq 5 is consistent with the $\sigma^2 / \tilde{\sigma}^2 = c_0 \log(N \tilde{\sigma})$ from the fitness wave theory? Fourth, you say "As shown in Methods, this parameter $[c]$ has a simple interpretation:..." I think a few sentences more here in the main text would help, c is the crucial quantity. Fifth, you say "The calibration between theory and data involves the

fitting of a single global amplitude c_0 "; going back to the above assertions about c_0 , can you give some intuition of what is the meaning of this parameter being fitted?

6) Can you comment on the sharpness of the transition in Fig 5a?

7) There is an interesting analogy that you might want to highlight between population genetic interference discussed in this paper and crosstalk interference (also leading to superlinear degradation of performance in signaling networks) discussed in Friedlander, Prizak et al, Nat Comms 7 (2016); in both cases, it is crucial to self-consistently consider the functioning / evolution of one component in a background of others that are structurally similar. A followup paper by the same authors (Nat Comms 8, 2017) is a further example of nontrivial evolution in sigmoidal biophysical fitness landscapes (refs 24-30 in your paper) of the same kind as the stability/affinity in your paper.

Gasper Tkacik

Reviewer #2 (Remarks to the Author):

This manuscript studies clonal interference dynamics in a model of quantitative biophysical traits, focusing particularly on how such dynamics may constrain the size of genomes. I consider the general problem interesting and their approach appealing. However, I did not gain many new insights from their approach, but rather found their results mainly technical. Therefore as it stands, I suspect the paper will be accessible only to specialists in theoretical population genetics who will be interested in the model. Please see below for lists of my general and specific comments.

General comments:

1. The paper's main novelty seems to be studying quantitative traits in the regime of clonal interference, thereby introducing the concept of "phenotypic interference." It was not clear to me what is gained from this approach beyond the existing body of work on clonal interference. The authors imply that "phenotypic interference" is a fundamentally new concept, but to me it just seems like clonal interference applied to quantitative traits. I furthermore found the emphasis on biophysical traits like protein stability misleading, because it didn't seem important to the authors' results: they didn't apply their results to any protein evolution data, nor did they make the argument that destabilizing mutations to proteins should be of dominant importance. Is the point just to use any nonlinear, diminishing-returns fitness function? If so, I think it would be clearer to consider abstract traits with such properties, making the argument that these fitness functions should be common among molecular traits (with protein stability being just one example).

2. The main result of this paper seems to be predicting how clonal interference of quantitative traits constrains genome size. This is of course an important problem, but there is a long history of studies addressing it, and I did not feel that literature was adequately discussed and compared to the present result. This made it hard for me to tell how significantly different the authors' results are. In particular, constraints on genome size resulting from accumulation of mutations destabilizing protein folding was discussed by Zeldovich et al. (PNAS 2007). As far as I can tell, this paper is cited in the Methods, but not discussed in the context of the results.

3. Overall the paper is quite technical and difficult to read. In particular, the main text contains quite a lot of notation (some of which I think is nonstandard and possibly unnecessary --- see specific comments below). The authors should consider significant simplifications to the presentation of their results if they hope for them to be accessible to an audience beyond theoretical population geneticists.

Specific comments:

1. It would be helpful for the authors to include line numbers in their manuscript --- this would make it easier to refer to specific items in the text. I will do my best below to indicate where in the text my comments apply.

2. In my opinion Fig. 1b comes too early in the text --- the introduction, at which point I was definitely not prepared to understand it. Shouldn't this plot be part of a main figure in the Results section, where it can be fully explained?

3. The colors in Fig. 2 are called "blue" and "orange," although for me they appeared to be more of a teal and red. I'm not sure if this is a difference in software/hardware or my own vision, but the authors might double-check these colors in case readers get confused.

4. The main text introduces a lot of new notation, some of which appears to be nonstandard and unnecessary. For example, they use G for the stability of a protein rather than the more standard ΔG , which is compounded by defining ΔG as the variance. Also, why use Γ instead of \bar{G} , and $\tilde{\sigma}$ instead of N_e^{-1} ? To me this made many of the results confusing, because these expressions fundamentally involve classical combinations of selection coefficients, mutation rates, and effective population sizes, but their notation obscures this intuition.

5. The authors make a lot of mathematical statements without adequate explanation. For example, the authors need to provide conceptual explanations for why $\Delta G f(\Gamma)$ is the rate of stability increase by selection, and where the mutation-selection balance condition ($\Delta G f(\Gamma) = u\epsilon_G$) comes from. Full derivations in the Methods/SI were also not cited for these.

6. In the section called "Evolution of a quantitative trait under interference selection," where is the effect of linkage? I couldn't tell the difference between these results and what one would expect for an unlinked trait evolving independently.

7. "The strength of selection on genetic variants is not fixed a priori, but is an emergent property of the global evolutionary process. A faster pace of evolution, i.e., an increase in coalescence rate $\tilde{\sigma}$, reduces the efficacy of selection." This sounds like restating a very standard concept in population genetics, that absolute selection strength doesn't matter so much as its value relative to competing processes like genetic drift (coalescence) and mutation. If the authors are implying something beyond this, then they should clarify.

8. "The non-adaptive, stationary scenario of housekeeping evolution..." I think calling that scenario "non-adaptive" is misleading: it is a steady state, so the mean fitness doesn't change, but there is still selection. To me "non-adaptive evolution" is devoid of any selection, e.g., purely genetic drift, mutation, recombination.

9. Above Eq. 3, the parameter c_0 appears and is given a value but with no explanation. Is it related to c in Eq. 5?

10. In general Eqs. 3, 4, 5 are presented without adequate explanation. I gather they define properties of the fitness wave, but the authors need to introduce fitness waves more systematically and explain how they are parameterized.

11. Figure 3 purports to show the two scaling regimes derived in the text between independently-evolving genes and the linkage-dominated case. However, the plots are not convincing to me that their analytical results are particularly accurate --- the data does not match either regime all that well, nor does it show a clear crossover. This is especially concerning given that the data is from

simulations, so whatever approximations the authors use in their analytical results may be problematic.

12. The text leading up to Eq. 6 should precisely define genetic load before introducing results on it.

13. At the bottom of page 6, the authors start mentioning the effect of all sorts of other scenarios on the genetic load result (Eq. 6), but this is way too terse to understand, and there was no reference to the SI which I later realized fully explains these cases.

14. I was surprised to see no mention of Neher and Shraiman (PNAS 2009) in the section on "The pathway to sexual evolution." I think that paper, which also shows the transition between linkage-dominated and recombination-dominated cases, is an important comparison to this manuscript's results.

15. "The spectrum of site selection coefficients is not a fixed input, but a dynamical output of the evolutionary process. That is the main difference of our approach to previous population-genetic models of asexual evolution [24-30]." As I understand it, this statement does not seem to be supported by the references or the preceding text. The citations mentioned here also have nonlinear fitness functions, where the strength of selection on a trait depends on the current value of the trait. The main difference, I would say, is a lack of clonal interference.

16. The parameterization of the fitness function in Eqs. 10 and 11 does not seem to allow for essential proteins or those that are otherwise lethal when unfolded. As long as f_0 is finite, then even $G \rightarrow -\infty$ for a single protein will not result in zero total fitness (Eq. 11).

Reviewer #1

In this paper, the authors build a theory of "phenotypic interference". In short, this is a theory for the evolution of clonal populations in which (even housekeeping) selection for many additive traits (e.g. thermodynamic stability of proteins) results in a load that scales quadratically with the number of traits due to clonal interference. This mode of evolution has been previously unobserved, because (i) of neglecting biophysical realism due to nonlinear behavior of individual traits; (ii) of nontrivial coupling between the local (single trait) and global (fitness wave) levels of evolution.

I believe that this paper makes a fundamental contribution to the field and recommend that it is published.

My major recommendation is that the clarity of initial exposition, as I describe below, could be increased to make the paper parseable more widely than to the specialized audience dealing with clonal interference. Several concrete suggestions:

We thank the referee for his general positive feedback and for his concrete criticism that helped us to significantly improve the manuscript. We revised the manuscript extensively to make it more comprehensible for a general audience, e.g., with Box 1. Moreover, we went over all derivations again to streamline them. You find specific points in the detailed response below.

1) For someone not familiar with the concept of "fitness wave" in detail it is hard to parse the intro para "Over a wide range of model parameters..." What I would find more important is to highlight first the feedback between local and global selection and explain what this is in words. Here, or perhaps later (in "theory of phenotypic interference" initial para) it would really help to highlight how such feedback emerges: through the fact that the $\tilde{\sigma}$ (related to eff. pop size) is set by the evolutionary dynamics itself. This is interesting to wider field as by default eff. pop size is often just assumed to be some constant, which would be wrong in this context. This is, if I understand properly, the gist of point (ii) above;

Thanks for these suggestions. We have now completely reorganized the introduction to highlight the feedback between global and local selection as the central new concept of the paper. The relevant paragraph now reads

Selection against genome complexity can be traced to a specific evolutionary mode of *phenotypic interference*, which is shown to occur in a parameter regime appropriate for typical microbial systems. The key characteristics of this mode are summarized in Box 1. First, local quantitative traits of a given gene are in an evolutionary equilibrium where the long-term average of the trait value, and its position of on the fitness landscape, are determined by the uphill force of selection and the downhill force of mutations (Box 1, top). Second, global genome evolution takes place in a so-called fitness wave; that is, genetic and phenotypic variants in multiple genes co-exist in a population and generate a broad distribution of fitness values [Desai and Fisher, 2007, Good et al., 2012, Hallatschek, 2011, Neher and Hallatschek, 2013, Neher et al., 2013, Rouzine et al., 2008] (Box 1, center). In a fitness wave, selection rather than genetic drift determines the coalescence rate of a population, which is the inverse of the effective population size [Neher et al., 2013, Rice et al., 2015]. Third, the defining feature of phenotypic interference is that global and local evolution are coupled in a specific way: the global coalescence rate sets the average selection coefficient of local trait changes. This feedback and its consequences are not covered by the existing models of fitness waves [Desai and Fisher, 2007, Good et al., 2012, Hallatschek, 2011, Neher and Hallatschek, 2013, Neher et al., 2013, Rouzine et al., 2008] and of quantitative trait evolution [Berg et al., 2004, Chen and Shakhnovich, 2009, Chi and Liberles, 2016, Gerland and Hwa, 2002, Goldstein, 2011, Manhart and Morozov, 2015, Nourmohammad et al., 2013, Serohijos and Shakhnovich, 2014, Zeldovich et al., 2007].

In addition, we have added a new Box 1 that summarizes the selective feedback and the key scales of selection and coalescence in a non-technical way. Box 1 also serves as a shortcut through the theory section for readers interested primarily in the biological results.

2) I would also highlight more point (i) above, namely the necessity of treating the essential biophysical nonlinearity (figure 1); right now, this is said in caption to Fig 1 (where you compare biophys model in red and discrete fitness model in blue), but I suggest you highlight this when you call out Figure 1B.

The importance of nonlinearities in gene fitness landscapes is now discussed in the introductory paragraph on biophysical models:

Selection on a gene is described by a thermodynamic fitness landscape $f(G)$, which is a sigmoid function with a high-fitness plateau corresponding to stable proteins and a low-fitness plateau corresponding to unfolded proteins (Box 1, top). We also discuss a stability-affinity protein model with a two-dimensional fitness landscape $f(G, E)$; this model includes enzymatic or regulatory functions of genes, specifically the protein binding affinity E to a molecular target. From the perspective of molecular evolution, these landscapes provide a biophysically grounded model of *local fitness epistasis*, which couples all sequence sites contributing to a stability or affinity trait in the same gene. Importantly, genome-wide local epistasis in protein-coding sequence operates independently on any assumptions of fitness interactions *across* genes. Beyond proteins, local epistasis occurs ubiquitously in quantitative molecular traits associated with binding interactions. This form of epistasis will be an important building block of our dynamical analysis but is not covered by the standard theory of asexual evolution [Desai and Fisher, 2007, Gerrish and Lenski, 1998, Good et al., 2012, Hallatschek, 2011, Neher and Hallatschek, 2013, Neher et al., 2013, Rouzine et al., 2008, Schiffls et al., 2011, Tsimring et al., 1996].

The role of nonlinearities and fitness epistasis is again taken up in the Discussion.

3) As above, for "Theory of phenotypic interference" first para, prepare the reader how the local and global fitness dynamics will be coupled, i.e., by mentioning specifically that the local part will assume some $\tilde{\sigma}$, which will be then set by the global fitness wave part. This I understood only after reading through the whole manuscript first and being confused.

To take this point into account, we have reorganized the entire theory section to make the logical structure clearer. The introductory theory paragraph now reads

The solution of the minimal model depends on a self-consistent closure of the local and global evolutionary dynamics, which are illustrated in Box 1. We will discuss these components in order. First, we study the evolutionary equilibrium of an individual biophysical trait in a process with given coalescence rate $\tilde{\sigma}$. Such traits evolve in a low-mutation regime characterized by strongly peaked trait distributions within a population and strong fluctuations over time and across genes (Supplementary Figure S1a). We show that the single-trait equilibrium depends in a simple way on the global parameter $\tilde{\sigma}$. Second, for the steady state of housekeeping evolution, the fitness variances of all traits combine to a global fitness wave (Supplementary Figure S1b). In tune with fitness wave theory, the wave pattern is of approximately Gaussian form and reflects broadly distributed, standing fitness variation maintained by mutations, selection, and coalescence [Desai and Fisher, 2007, Good et al., 2012, Hallatschek, 2011, Neher and Hallatschek, 2013, Neher et al., 2013, Rouzine et al., 2008]. The global fitness wave and, hence, the coalescence rate $\tilde{\sigma}$, are determined by the ensemble of traits evolving under genetic linkage. Conversely, $\tilde{\sigma}$ determines the single-trait equilibria, leading to a closure of the derivation.

4) Minor, in the subsection "Evolution of a quantitative trait under interference selection" – that subsection is just a general section on the evolution of a quantitative trait, as the interference doesn't come in yet, correct? Perhaps you can amend the title.

Also: can you make clear what parts of this subsection are novel (e.g., relations in Eq (1) and (2) appear straightforward, has this been done before)?

We agree that the scaling relations of an individual trait as a function of the coalescence rate $\tilde{\sigma} = N_e^{-1}$ hold regardless if this quantity is set by genetic drift (as is the case for recombining populations) or by selection (which applies to fitness waves). We now state this explicitly:

In a fitness wave, the parameter $\tilde{\sigma}$ couples each individual trait to the global evolutionary dynamics of all genetically linked genes (Box 1). In contrast, an independently evolving trait depends on an effective population size N_e set by genetic drift.

The specifics of interference selection enter through the derivation of the diversity formula (6), which is given in Methods. It turns out that the “effectively neutral” form of the diversity applies throughout the phenotypic interference regime even for traits with many sequence sites, while it would be violated in large populations governed by genetic drift.

We also agree that the calculations in this section are quite straightforward and extend previous results for neutral sequence diversity [Gillespie, 2000, Good et al., 2014, Kimura, 1983, Rice et al., 2015] and for quantitative traits under genetic drift [Charlesworth, 2013, Keightley and Hill, 1988, Lynch and Hill, 1986, Nourmohammad et al., 2013], as stated in the text. The new part is the derivation of eqs. (1), (3) for traits *under selection* in the phenotypic interference regime; see eqs. (16) – (17). Because interference selection is quite different from genetic drift, this is not clear a priori, and it is an important ingredient for the entire self-consistent solution. We have rewritten this section to make the flow of arguments clearer.

5) I had the most trouble parsing the crucial section “Housekeeping evolution of multiple traits”. First, it would be good to be more clear what are the necessary conditions for the fitness wave regime to hold in this case. Make it more explicit that the fitness wave theory for your case contributes the crucial equation that relates the fitness variance and the coalescence rate, and perhaps give a few sentence intuition behind it.

We have rewritten the text according to your suggestion:

In the housekeeping state, the total fitness variance σ^2 is simply the sum of the fitness variances Δ_f of the individual genes (Supplementary Figure S4). Using Equation (3), this sum rule takes the form $\sigma^2 = g\Delta_f = 2ug\tilde{\sigma}$, which relates the scales of global selection and coalescence, σ and $\tilde{\sigma}$. Given a sufficient supply of non-neutral mutations, global evolution proceeds in a fitness wave (the condition for wave occurrence will be made precise below). General fitness wave theory then provides a second relation between global selection and coalescence,

$$c \equiv \frac{\sigma^2}{\tilde{\sigma}^2} = c_0 \log(N\sigma), \quad (1)$$

where N is the population size and c_0 is a model-dependent prefactor [Neher and Hallatschek, 2013, Neher et al., 2013] (Methods). Combining these relations, we obtain the global fitness wave of phenotypic interference, ...

We also added a new Methods section “Housekeeping equilibrium and fitness waves” that provides some background on fitness waves including short derivations, to make the text more self-contained.

Moreover, currently you assert $c_0 \sim 100$, which requires at least a few words about where the magic constant emerges from.

The prefactor c_0 in eq. (4) is model-dependent and known only in asymptotic cases, which is now explained in Methods, “Housekeeping equilibrium and fitness waves”. For this reason, we treat c_0 as a fit parameter; this has been clarified in the main text and in Fig. 1.

Second, you refer to c as the “slowly varying parameter”. In what sense is this slow, can you be more precise?

This refers to the logarithmic dependence of c on evolutionary parameters. We have made this explicit in eq. (7) and in Methods, “Housekeeping equilibrium and fitness waves”.

Third, can you comment briefly about how Eq 5 is consistent with the $\sigma^2/\tilde{\sigma}^2 = c_0 \log(N\sigma)$ from the fitness wave theory?

The step is now made explicit in the main text, from eq. (4) to eq. (7), and is further explained in Methods. It involves a series expansion. From (4), we have

$$c = \sigma^2/\tilde{\sigma}^2 = c_0 \log(N\sigma) = c_0 \log(2Nug/c).$$

This equation can be solved analytically, $c = \frac{1}{2}c_0 \text{ProdLog}(8(Ngu)^2/c_0)$, which is for large Ngu

$$c = c_0 \log(2\sqrt{2}Nug/c_0) + \log \log(2\sqrt{2}Nug/c_0).$$

Fourth, you say "As shown in Methods, this parameter [c] has a simple interpretation:..." I think a few sentences more here in the main text would help, c is the crucial quantity.

Thanks, we have added this to the main text below eq. (7), to Box 1, and to Methods, "Housekeeping equilibrium and fitness waves". The parameter c is indeed ubiquitous: in any fitness wave, it relates both the coalescence time and the total fitness span to the fitness variance (Box 1 and Methods). Under phenotypic interference, it is related in a specific way to the wave complexity (Box 1 and eq. (19)). In a nutshell, c measures how many mutational steps (of size $\sim \tilde{\sigma}$) are required to cover one fitness span $\tilde{\sigma}$.

Fifth, you say "The calibration between theory and data involves the fitting of a single global amplitude c_0 "; going back to the above assertions about c_0 , can you give some intuition of what is the meaning of this parameter being fitted?

As discussed above, c_0 is a model-dependent prefactor in the general fitness wave relation (4). Therefore, we treat it as a global fit parameter, which is now explicitly quoted in Fig. 1.

6) Can you comment on the sharpness of the transition in Fig 5a?

The derivation in the section "The pathway to sexual evolution", which predicts a first order transition, is based on infinite system size. In a finite system, first order transitions are always rounded. In the present system, there are two sources of finite-size scaling: the number of genes g , and the population size N . As expected, the simulation results are rounded consistently for different gene numbers. We now comment on this directly in Fig. 4a.

7) There is an interesting analogy that you might want to highlight between population genetic interference discussed in this paper and crosstalk interference (also leading to superlinear degradation of performance in signaling networks) discussed in Friedlander, Prizak et al, Nat Comms 7 (2016); in both cases, it is crucial to self-consistently consider the functioning / evolution of one component in a background of others that are structurally similar. A followup paper by the same authors (Nat Comms 8, 2017) is a further example of nontrivial evolution in sigmoidal biophysical fitness landscapes (refs 24-30 in your paper) of the same kind as the stability/affinity in your paper.

Thanks for pointing us to these papers. Indeed the analogy to [Friedlander et al., 2016] is very interesting and we mention the paper in the discussion. Moreover, we list the fitness landscape of [Friedlander et al., 2017] as another example for the model of the fitness landscape.

Reviewer #2

This manuscript studies clonal interference dynamics in a model of quantitative biophysical traits, focusing particularly on how such dynamics may constrain the size of genomes. I consider the general problem interesting and their approach appealing. However, I did not gain many new insights from their approach, but rather found their results mainly technical. Therefore as it stands, I suspect the paper will be accessible only to specialists in theoretical population genetics who will be interested in the model. Please see below for lists of my general and specific comments.

We thank the reviewer for assessing our manuscript and giving valuable feedback. In response, we have rewritten the paper, and we hope the revised version highlights the biological results in a better way. In the Discussion, we recap four novel points: the new picture of interference selection at the phenotypic level, the systems-biological implications for evolution experiments and for macro-evolution, and the specific pathway to recombination discussed in the paper. Throughout the paper, the role of the underlying biophysics for these results is explained better.

General comments:

1. The paper's main novelty seems to be studying quantitative traits in the regime of clonal interference, thereby introducing the concept of "phenotypic interference." It was not clear to me what is gained from

this approach beyond the existing body of work on clonal interference. The authors imply that "phenotypic interference" is a fundamentally new concept, but to me it just seems like clonal interference applied to quantitative traits.

We now address this point comprehensively in several places of the manuscript. This paper is the first to combine fitness wave theory and the theory of quantitative traits. While asexual evolution in global fitness waves and evolution on biophysical fitness landscapes (e.g., for protein stability) are individually well-studied subjects, the merger of these into a consistent model produces qualitatively new results. The main reason is the distinct selective forces that drive the fitness wave in our model: we use mechanistically grounded biophysical fitness landscapes instead of ad hoc assumptions on the spectrum of site selection coefficients. This is now made explicit already in the introduction:

From the perspective of molecular evolution, these landscapes provide a biophysically grounded model of *local fitness epistasis*, which couples all sequence sites contributing to a stability or affinity trait in the same gene. Importantly, genome-wide local epistasis in protein-coding sequence operates independently on any assumptions of fitness interactions *across* genes. Beyond proteins, local epistasis occurs ubiquitously in quantitative molecular traits associated with binding interactions. This form of epistasis will be an important building block of our dynamical analysis but is not covered by the standard theory of asexual evolution [Desai and Fisher, 2007, Gerrish and Lenski, 1998, Good et al., 2012, Hallatschek, 2011, Neher and Hallatschek, 2013, Neher et al., 2013, Rouzine et al., 2008, Schiffels et al., 2011, Tsimring et al., 1996].

Selection against genome complexity can be traced to a specific evolutionary mode of *phenotypic interference*, which is shown to occur in a parameter regime appropriate for typical microbial systems. The key characteristics of this mode are summarized in Box 1. First, local quantitative traits of a given gene are in an evolutionary equilibrium where the long-term average of the trait value, and its position on the fitness landscape, are determined by the uphill force of selection and the downhill force of mutations (Box 1, top). Second, global genome evolution takes place in a so-called fitness wave; that is, genetic and phenotypic variants in multiple genes co-exist in a population and generate a broad distribution of fitness values [Desai and Fisher, 2007, Good et al., 2012, Hallatschek, 2011, Neher and Hallatschek, 2013, Neher et al., 2013, Rouzine et al., 2008] (Box 1, center). In a fitness wave, selection rather than genetic drift determines the coalescence rate of a population, which is the inverse of the effective population size [Neher et al., 2013, Rice et al., 2015]. Third, the defining feature of phenotypic interference is that global and local evolution are coupled in a specific way: the global coalescence rate sets the average selection coefficient of local trait changes. This feedback and its consequences are not covered by the existing models of fitness waves [Desai and Fisher, 2007, Good et al., 2012, Hallatschek, 2011, Neher and Hallatschek, 2013, Neher et al., 2013, Rouzine et al., 2008] and of quantitative trait evolution [Berg et al., 2004, Chen and Shakhnovich, 2009, Chi and Liberles, 2016, Gerland and Hwa, 2002, Goldstein, 2011, Manhart and Morozov, 2015, Nourmohammad et al., 2013, Serohijos and Shakhnovich, 2014, Zeldovich et al., 2007].

In addition, Box 1 summarizes the salient features of phenotypic interference in a non-technical way.

The new results are now succinctly captured in the discussion. In particular, the nonlinear scaling of the load and its implications, as well as the specific pathway for the evolution of sex, have not been reported before. This pathway builds again on mechanistically grounded epistasis; its differences to previous models based on fitness waves [Neher, 2013, Neher et al., 2013, Weissman and Barton, 2012, Weissman and Hallatschek, 2014] and to models based on *global* epistasis are now clearly described.

I furthermore found the emphasis on biophysical traits like protein stability misleading, because it didn't seem important to the authors' results: they didn't apply their results to any protein evolution data, nor did they make the argument that destabilizing mutations to proteins should be of dominant importance. Is the point just to use any nonlinear, diminishing-returns fitness function? If so, I think it would be clearer

to consider abstract traits with such properties, making the argument that these fitness functions should be common among molecular traits (with protein stability being just one example).

Thank you for this comment, which has been instrumental for the revision. We now explain the role of biophysics in a better way throughout the manuscript. In a nutshell, this role is not just to give examples of molecular quantitative traits, but also to provide mechanistically grounded models of epistatic selection. A cell does have about 5000 proteins, so the genome-wide selection map does include about 5000 local fitness landscapes of sigmoid form for fold stability alone. As we show in this paper, the ensemble of these landscapes leads to different evolutionary regimes than previous models of asexual evolution based on abstract assumptions on selection. Without the biophysical underpinning, our epistatic model would be just as good as any other assumption.

Regarding the specifics of protein evolution, we use experimental data in our model, largely informed by refs. [Chen and Shakhnovich, 2009, Serohijos and Shakhnovich, 2014, Zeldovich et al., 2007]. Evidence for the pervasive role of protein stability in molecular evolution has come from a number of recent experimental works. We discuss in detail one example, the analysis of Lenski’s long-term experiment, in light of our model [Couce et al., 2017, Good et al., 2017, Leiby and Marx, 2014, Tenaillon et al., 2016]. But the thrust of the macro-evolutionary part of the paper is indeed somewhat different: instead of arguing for the importance of protein evolution compared to other molecular units, we show that already a minimal model with only housekeeping evolution of proteins has strong implications for genome complexity. More comprehensive models with more molecules and with ecological adaptations, which doubtlessly are relevant in parallel to protein stability evolution, would only increase these effects. But even the minimal model needs a biophysical underpinning for credibility.

2. The main result of this paper seems to be predicting how clonal interference of quantitative traits constrains genome size. This is of course an important problem, but there is a long history of studies addressing it, and I did not feel that literature was adequately discussed and compared to the present result. This made it hard for me to tell how significantly different the authors’ results are. In particular, constraints on genome size resulting from accumulation of mutations destabilizing protein folding was discussed by Zeldovich et al. (PNAS 2007). As far as I can tell, this paper is cited in the Methods, but not discussed in the context of the results.

The paper by [Zeldovich et al., 2007] and follow-up papers are seminal works that informed much of our thinking about protein evolution. We believe we have carefully cited this group of papers throughout the manuscript; the relation between [Zeldovich et al., 2007] and our work is now discussed at several places of the main text. In a nutshell, we use exactly the same “mutational wind” as [Zeldovich et al., 2007] but there are two main differences between [Zeldovich et al., 2007] and our work¹:

(i) We treat the evolution of genetically linked proteins under interference selection, while [Zeldovich et al., 2007] assume factorization across genes, which amounts to free recombination: “Because mutations in all proteins are independent, and boundary conditions are the same in each dimension (i.e., for each gene), one can write $P(E) = \prod_{i=1}^{\Gamma} p(E_i)$, separate the variables, and reduce the problem to a product of Γ 1D diffusion problems for each gene: [...]” [Zeldovich et al., 2007]. Their paper mentions possible deviations from factorization at higher mutation rates but does not treat interference selection between these mutations.

(ii) We show that in the phenotypic interference regime, individual traits evolve in a monomorphic low-mutation regime, while [Zeldovich et al., 2007] assume a highly polymorphic quasispecies (the solution of an imaginary-time Schrödinger equation) for each trait (the regime $Nm > 1$ in their notation, see also reply to point 6).

These differences translate into qualitatively and quantitatively different results. Linked evolution of quantitative traits produces a superlinear load, $\mathcal{L} \sim ug^2$ (data and red line in Fig. 2); [Zeldovich et al., 2007] find a linear load, $L \sim ug$, which we recover for recombining populations (brown line in Fig. 2). The difference between these can be orders of magnitude.

3. Overall the paper is quite technical and difficult to read. In particular, the main text contains quite a lot of notation (some of which I think is nonstandard and possibly unnecessary — see specific comments below).

¹Notation translation: $m = \mu, \Gamma = g, b = f_0, h = \epsilon$, and $E_i = -G_i$ are the traits.

The authors should consider significant simplifications to the presentation of their results if they hope for them to be accessible to an audience beyond theoretical population geneticists.

We addressed this comment throughout the paper. In particular, Box 1 provides a non-technical summary of phenotypic interference aimed at a broad readership. Throughout the main text, we try to keep technicalities at a minimum, while the Methods section now provides more background to make the paper self-contained. The biological implications section can now be read with just Box 1 as background.

Specific comments:

1. It would be helpful for the authors to include line numbers in their manuscript — this would make it easier to refer to specific items in the text. I will do my best below to indicate where in the text my comments apply.

We are sorry for the inconvenience and added line numbers to the new version.

2. In my opinion Fig. 1b comes too early in the text — the introduction, at which point I was definitely not prepared to understand it. Shouldn't this plot be part of a main figure in the Results section, where it can be fully explained?

In the revised version, this figure has been trimmed and is now a schematic within Box 1. We think it is important to give this take-home message at an early point in the manuscript. All quantitative analysis is in the later figures, which come in parallel with the detailed explanation.

3. The colors in Fig. 2 are called "blue" and "orange," although for me they appeared to be more of a teal and red. I'm not sure if this is a difference in software/hardware or my own vision, but the authors might double-check these colors in case readers get confused.

Thanks, we will make sure that the final version of figures has unambiguous colors.

4. The main text introduces a lot of new notation, some of which appears to be nonstandard and unnecessary. For example, they use G for the stability of a protein rather than the more standard ΔG , which is compounded by defining Δ_G as the variance. Also, why use introduce Γ instead of \bar{G} , and $\bar{\sigma}$ instead of N_e^{-1} ? To me this made many of the results confusing, because these expressions fundamentally involve classical combinations of selection coefficients, mutation rates, and effective population sizes, but their notation obscures this intuition.

We are well aware of other notations and their conflicts, especially with the community of protein evolution. However, there is a trade-off between the notations between scientific communities: ΔG would collide with the Δ widely used for diversities in population genetics. For example, $\Delta_{\Delta G}$ from population genetics collides with $\Delta\Delta G$ for mutational effects in protein evolution. Denoting the mean trait Γ by \bar{G} can again cause confusion, for example in the probability distribution $Q(\Gamma)$, which is the cross-species distribution of the within-species mean trait. Similarly, $\bar{\sigma} = N_e^{-1}$ is the inverse of the effective population size, but it is fundamentally a selection scale and not related to genetic drift as the classical notation N_e may suggest. To emphasize the links between selection scales, e.g. in Box 1, we prefer to use a notation as introduced in [Schiffels et al., Genetics 2011; Rice et al., Genetics 2015]. However, we added more explanatory text and alternative notations throughout the text.

5. The authors make a lot of mathematical statements without adequate explanation. For example, the authors need to provide conceptual explanations for why $\Delta_G f'(\Gamma)$ is the rate of stability increase by selection, and where the mutation-selection balance condition ($\Delta_G f'(\Gamma) = u\epsilon_G$) comes from. Full derivations in the Methods/SI were also not cited for these.

There is a trade-off between readability and level of detail in the main text. To address your comment, we added several more detailed derivations to Methods, including a careful derivation of the mutation-selection balance condition ($\Delta_G f'(\Gamma) = u\epsilon_G$) in the section "Evolutionary model". We also linked the explanations in Methods to the main text in a better way.

6. In the section called "Evolution of a quantitative trait under interference selection," where is the effect of linkage? I couldn't tell the difference between these results and what one would expect for an unlinked trait evolving independently.

We agree that the relations (1) - (3) would formally take the same form in standard population genetics, if we use the identification $N_e = \tilde{\sigma}^{-1}$. This is now clearly stated in the text:

This form extends previous results on neutral sequence diversity [Gillespie, 2000, Good et al., 2014, Kimura, 1983, Nourmohammad et al., 2013, Rice et al., 2015] and on traits under genetic drift [Keightley and Hill, 1988, Lynch and Hill, 1986, Nourmohammad et al., 2013].

The question is whether this form is actually valid for traits under diminishing-return selection and genetic linkage. This is far from obvious and has, to our knowledge, not been established before. In Methods, we show that the form (1) for the diversity is self-consistently valid in the phenotypic interference regime. Here is where the specifics of $\tilde{\sigma}$ as a selection scale independent of genetic drift come in. In contrast, $\Delta_G = 2N_e u \epsilon_G^2$ would often cease to be valid for complex traits evolving independently. This is the case, for example, in the regime of [Zeldovich et al., 2007], where Δ_G is shaped by selection (in the notation of [Zeldovich et al., 2007], the trait variance is the width of the distribution $p(\Delta G)$ in their eq. [6a]). Our main text states this:

In Methods, we derive Equation (1 for quantitative traits in a fitness landscape $f(G)$ by showing that stabilizing selection on Δ_G can be neglected throughout the phenotypic interference regime; this scaling is confirmed by simulations (Supplementary Figure S2a). In a fitness wave, the parameter $\tilde{\sigma}$ couples each individual trait to the global evolutionary dynamics of all genetically linked genes (Box 1). In contrast, an independently evolving trait depends on an effective population size N_e set by genetic drift.

7. "The strength of selection on genetic variants is not fixed a priori, but is an emergent property of the global evolutionary process. A faster pace of evolution, i.e., an increase in coalescence rate $\tilde{\sigma}$, reduces the efficacy of selection." This sounds like restating a very standard concept in population genetics, that absolute selection strength doesn't matter so much as its value relative to competing processes like genetic drift (coalescence) and mutation. If the authors are implying something beyond this, then they should clarify.

The statement does say more than standard population genetics. The standard concept you refer to is that selection coefficients enter fixation probabilities via the product with the effective population size, $2N_e s = s/\tilde{\sigma}$. Eq. (1) states that this product is tuned to a fixed average value,

$$2N_e s = 2.$$

The text below eq. (1) gives a non-technical explanation of this result. We have rewritten this text to make it clear that the standard concept is just one part of the argument.

8. "The non-adaptive, stationary scenario of housekeeping evolution..." I think calling that scenario "non-adaptive" is misleading: it is a steady state, so the mean fitness doesn't change, but there is still selection. To me "non-adaptive evolution" is devoid of any selection, e.g., purely genetic drift, mutation, recombination.

There are different definitions of adaptation in the literature. We clarified in revised the text that we use a standard definition of evolutionary and population biology; see e.g., [McDonald and Kreitman, 1991, Orr, 2005]. Here adaptation is defined by a net surplus of beneficial mutations and is clearly distinguished from evolutionary equilibrium. According to that definition, the housekeeping state has no adaptation because the average protein stability remains constant in time.

9. Above Eq. 3, the parameter c_0 appears and is given a value but with no explanation. Is it related to c in Eq. 5?

c_0 is a model-dependent prefactor in the fitness wave relation (4). We have clarified the main text of this section; in Methods "Housekeeping equilibrium and fitness waves", we provide more background on eq. (4).

10. In general Eqs. 3, 4, 5 are presented without adequate explanation. I gather they define properties of the fitness wave, but the authors need to introduce fitness waves more systematically and explain how they are parameterized.

Thanks for this suggestion. In response, the section "Housekeeping evolution of multiple traits" has been completely rewritten to improve clarity. In addition, the key fitness wave relation (4) is explained in Methods, "Housekeeping equilibrium and fitness waves". Box 1 gives a non-technical summary of all this.

11. Figure 3 purports to show the two scaling regimes derived in the text between independently-evolving genes and the linkage-dominated case. However, the plots are not convincing to me that their analytical results are particularly accurate — the data does not match either regime all that well, nor does it show a clear crossover. This is especially concerning given that the data is from simulations, so whatever approximations the authors use in their analytical results may be problematic.

Figure 1 (which was Fig. 3 before) shows the data (dots) and two theoretical scaling regimes: independently evolving genes (brown) and genes under phenotypic interference (red). We are not sure if we understand your comment since the data do interpolate between these theoretical curves. In particular, there is little doubt that the data show strong deviations from the brown line of independently evolving genes. The same is true for the genomic load in Fig. 2, which deviates by 2 orders of magnitude from the linear expectation (brown line) and is well approximated by a quadratic form. So the main point of the paper, the superlinear form of the genetic load, is unambiguously supported by Fig. 1 and Fig. 2.

Moreover, there is a number of important consistency checks: (a) In line with theory, the data decouple well from the selection amplitude f_0 of the gene. This has been tested for 4 orders of magnitude of selection coefficients (color code). (b) The crossover point $g_0 \sim c$ is shown consistently in all panels (red dashed line). (c) Even relative amplitudes are reproduced consistently over all panels of Fig. 1 and Fig. 2, there is only one global fit parameter c_0 (see caption of Fig. 1).

12. The text leading up to Eq. 6 should precisely define genetic load before introducing results on it.

Thanks, we now carefully define the genetic load:

The evolutionary cost of deleterious mutations can be quantified by the genetic load, which is defined as the mean fitness of a population compared to the fitness maximum. In the biophysical fitness landscape $f(G)$ of the minimal model, the load of a given gene takes the approximate form $f_0 - f(\Gamma)$, where Γ denotes the population mean stability and f_0 is the fitness of a fully stable gene ($G \gg 0$); see Box 1 (top) and Equation (13) in Methods.

13. At the bottom of page 6, the authors start mentioning the effect of all sorts of other scenarios on the genetic load result (Eq. 6), but this is way too terse to understand, and there was no reference to the SI which I later realized fully explains these cases.

We have extended this paragraph and improved the connection to the full explanations in SI:

In Supplementary Notes and Supplementary Figure S5, we discuss phenotypic interference in a number of extended biophysical models. These include active protein degradation at the cellular level, a ubiquitous process that drives the thermodynamics of folding out of equilibrium [Hochstrasser, 1996]. We show that thermodynamic non-equilibrium influences the fitness landscape but does not affect our qualitative and quantitative conclusions. Another example is the stability-affinity model, which has two quantitative traits per gene that evolve in a two-dimensional sigmoid fitness landscape $f(G, E)$ with both traits under (correlated) stabilizing selection [Chéron et al., 2016, Manhart and Morozov, 2015]. We show that under reasonable biophysical assumptions, evolution in a stability-affinity model produces a 2-fold higher interference load than the minimal model, $\mathcal{L}_{\text{int}}(g) \approx 8ug^2/c$. Alternative models with a quadratic single-peak fitness landscape describe, for example, gene expression levels under stabilizing selection [Nourmohammad et al., 2017]. We show that such landscapes generate an even stronger nonlinearity of the load, $\mathcal{L}_{\text{int}}(g) \sim g^3$. In contrast, a discrete model with a fitness effect f_0 of each gene shows a linear load up to a characteristic gene number $g_m = (f_0/u) \log(Nf_0)$ associated with the onset of mutational meltdown by Muller’s ratchet [Gordo and Charlesworth, 2000, Muller, 1964, Rouzine et al., 2008]. These examples suggest that superlinear scaling of the genetic load holds under quite general conditions, given a sufficient number of quantitative traits evolving under genetic linkage and in fitness landscapes with negative epistasis. This type of fitness landscape is ubiquitous in biophysical models.

14. I was surprised to see no mention of Neher and Shraiman (PNAS 2009) in the section on "The pathway to sexual evolution." I think that paper, which also shows the transition between linkage-dominated and recombination-dominated cases, is an important comparison to this manuscript's results.

Thank you for pointing out this reference. It gives another example for a crossover from selection on genotypes towards selection on genes; related crossovers are treated in the cited refs. [Neher, 2013, Neher et al., 2013]. Model and results of [Neher and Shraiman, 2009] clearly differ from ours. While we discuss a long-term evolutionary mode with the competition of trait mutations, [Neher and Shraiman, 2009] discuss the loss of diversity from an initial standing variation. Therefore, no new variants arise there and they describe short-term dynamics. This scenario cannot describe the evolution of recombination. Moreover, they show for the short-term fixation that small recombination rates may bring a small benefit, but the critical transition towards selection on genes instead of genotypes comes with a cost, see Fig. 4b of [Neher and Shraiman, 2009]. This is a property of heterogeneous *global epistasis*, which acts very differently from the local epistasis of biophysical fitness landscapes. We added a reference to [Neher and Shraiman, 2009] and a comparative discussion to the section "Comparison of models for the evolution of recombination":

In another model with broad, heterogeneous pairwise epistasis, recombination generates a transition from genotype selection to gene selection [Neher and Shraiman, 2009]. This causes a short-term benefit, but the transition causes a net fitness loss by breaking linkage between epistatic sites [Neher and Shraiman, 2009]. Our model replaces the assumption of global epistasis across the genome by local diminishing-return epistasis between loci affecting the same protein stability or affinity trait. This form of epistasis follows directly from the underlying thermodynamic nonlinearities, and its evolutionary effects differ from global epistasis.

15. "The spectrum of site selection coefficients is not a fixed input, but a dynamical output of the evolutionary process. That is the main difference of our approach to previous population-genetic models of asexual evolution [24-30]." As I understand it, this statement does not seem to be supported by the references or the preceding text. The citations mentioned here also have nonlinear fitness functions, where the strength of selection on a trait depends on the current value of the trait. The main difference, I would say, is a lack of clonal interference.

Sorry, there was a typo which has been fixed now. Of course, we mean the population genetic models of asexual evolution, which are the citations [Desai and Fisher, 2007, Gerrish and Lenski, 1998, Good et al., 2012, Hallatschek, 2011, Neher and Hallatschek, 2013, Neher et al., 2013, Rouzine et al., 2008, Schiffels et al., 2011, Tsimring et al., 1996].

16. The parameterization of the fitness function in Eqs. 10 and 11 does not seem to allow for essential proteins or those that are otherwise lethal when unfolded. As long as f_0 is finite, then even $G \rightarrow -\infty$ for a single protein will not result in zero total fitness (Eq. 11).

We added a comment to Methods on how our approach takes into account essential genes. As introduced in [Chen and Shakhnovich, 2009, Zeldovich et al., 2007], essential genes (for which lack of folding is lethal) can be modeled by a sharp cutoff of the fitness landscape. That is, our Malthusian fitness landscape $f(G)$ acquires a singularity in the regime of marginal folding ($f \rightarrow -\infty$ for $G \rightarrow G_0 \sim 0$). Importantly, however, the landscape $f(G)$ retains the form (13) in the regime of stable folding ($G/k_{BT} \gtrsim 1$), which implies that our conclusions remain unaffected. This is also consistent with our results on gene loss, Equation (10) and Fig. 3. We show that loss affects weakly selected genes, while essential genes are protected from loss by strong selection.

References

J. Berg, S. Willmann, and M. Lässig. Adaptive evolution of transcription factor binding sites. *BMC Evol Biol*, 4(1):42, 2004. ISSN 1471-2148. doi: 10.1186/1471-2148-4-42. URL <http://www.biomedcentral.com/1471-2148/4/42>.

- B. Charlesworth. Stabilizing selection, purifying selection, and mutational bias in finite populations. *Genetics*, 194(4):955–971, 2013. ISSN 0016-6731. doi: 10.1534/genetics.113.151555. URL <http://www.genetics.org/content/194/4/955>.
- P. Chen and E. I. Shakhnovich. Lethal mutagenesis in viruses and bacteria. *Genetics*, 183(2):639–650, 2009. doi: 10.1534/genetics.109.106492. URL <http://www.genetics.org/content/183/2/639.abstract>.
- N. Chéron, A. W. R. Serohijos, J.-M. Choi, and E. I. Shakhnovich. Evolutionary dynamics of viral escape under antibodies stress: A biophysical model. *Protein Sci*, 25(7):1332–1340, 2016. ISSN 1469-896X. doi: 10.1002/pro.2915. URL <http://dx.doi.org/10.1002/pro.2915>.
- P. B. Chi and D. A. Liberles. Selection on protein structure, interaction, and sequence. *Protein Sci*, 25(7):1168–1178, 2016. ISSN 1469-896X. doi: 10.1002/pro.2886. URL <http://dx.doi.org/10.1002/pro.2886>.
- A. Couce, L. V. Caudwell, C. Feinauer, T. Hindré, J.-P. Feugeas, M. Weigt, R. E. Lenski, D. Schneider, and O. Tenaillon. Mutator genomes decay, despite sustained fitness gains, in a long-term experiment with bacteria. *Proc Natl Acad Sci*, 114(43):E9026–E9035, 2017. doi: 10.1073/pnas.1705887114. URL <http://www.pnas.org/content/114/43/E9026.abstract>.
- M. M. Desai and D. S. Fisher. Beneficial mutation–selection balance and the effect of linkage on positive selection. *Genetics*, 176(3):1759–1798, 2007. doi: 10.1534/genetics.106.067678. URL <http://www.genetics.org/content/176/3/1759.abstract>.
- T. Friedlander, R. Prizak, C. C. Guet, N. H. Barton, and G. Tkačik. Intrinsic limits to gene regulation by global crosstalk. *Nature Communications*, 7:12307 EP –, 08 2016. URL <https://doi.org/10.1038/ncomms12307>.
- T. Friedlander, R. Prizak, N. H. Barton, and G. Tkačik. Evolution of new regulatory functions on biophysically realistic fitness landscapes. *Nat Commun*, 8(1):216, Aug. 2017. ISSN 2041-1723. URL <https://doi.org/10.1038/s41467-017-00238-8>.
- U. Gerland and T. Hwa. On the selection and evolution of regulatory DNA motifs. *J Mol Evol*, 55(4):386–400, 2002. doi: 10.1007/s00239-002-2335-z.
- P. J. Gerrish and R. E. Lenski. The fate of competing beneficial mutations in an asexual population. *Genetica*, 102:127–144, Mar 1998. ISSN 1573-6857. doi: 10.1023/A:1017067816551. URL <https://doi.org/10.1023/A:1017067816551>.
- J. H. Gillespie. Genetic drift in an infinite population: The pseudohitchhiking model. *Genetics*, 155(2):909–919, 2000. ISSN 0016-6731. URL <http://www.genetics.org/content/155/2/909>.
- R. A. Goldstein. The evolution and evolutionary consequences of marginal thermostability in proteins. *Proteins: Struct , Funct , Bioinf*, 79(5):1396–1407, 2011. ISSN 1097-0134. doi: 10.1002/prot.22964. URL <http://dx.doi.org/10.1002/prot.22964>.
- B. H. Good, I. M. Rouzine, D. J. Balick, O. Hallatschek, and M. M. Desai. Distribution of fixed beneficial mutations and the rate of adaptation in asexual populations. *Proc Natl Acad Sci*, 109(13):4950–4955, 2012. doi: 10.1073/pnas.1119910109. URL <http://www.pnas.org/content/109/13/4950.abstract>.
- B. H. Good, A. M. Walczak, R. A. Neher, and M. M. Desai. Genetic diversity in the interference selection limit. *PLoS Genet*, 10(3):e1004222, 03 2014. doi: 10.1371/journal.pgen.1004222. URL <http://dx.doi.org/10.1371%2Fjournal.pgen.1004222>.
- B. H. Good, M. J. McDonald, J. E. Barrick, R. E. Lenski, and M. M. Desai. The dynamics of molecular evolution over 60,000 generations. *Nature*, 551:45 EP –, 10 2017. URL <http://dx.doi.org/10.1038/nature24287>.

- I. Gordo and B. Charlesworth. The degeneration of asexual haploid populations and the speed of Muller's ratchet. *Genetics*, 154(3):1379–1387, 03 2000. URL <http://www.ncbi.nlm.nih.gov/pmc/articles/PMC1460994/>.
- O. Hallatschek. The noisy edge of traveling waves. *Proc Natl Acad Sci*, 108(5):1783–1787, 2011. doi: 10.1073/pnas.1013529108. URL <http://www.pnas.org/content/108/5/1783.abstract>.
- M. Hochstrasser. Ubiquitin-dependent protein degradation. *Annu Rev Genet*, 30:405–439, 1996. ISSN 0066-4197. doi: 10.1146/annurev.genet.30.1.405.
- P. D. Keightley and W. G. Hill. Quantitative genetic variability maintained by mutation-stabilizing selection balance in finite populations. *Genet Res*, 52(1):3343, 1988. doi: 10.1017/S0016672300027282.
- M. Kimura. *The Neutral Theory of Molecular Evolution*. Cambridge University Press, 1983. doi: 10.1017/CBO9780511623486.
- N. Leiby and C. J. Marx. Metabolic erosion primarily through mutation accumulation, and not tradeoffs, drives limited evolution of substrate specificity in escherichia coli. *PLOS Biology*, 12(2):1–10, 02 2014. doi: 10.1371/journal.pbio.1001789. URL <https://doi.org/10.1371/journal.pbio.1001789>.
- M. Lynch and W. G. Hill. Phenotypic evolution by neutral mutation. *Evolution*, 40(5):915–935, 1986. ISSN 00143820, 15585646. doi: 10.2307/2408753. URL <http://www.jstor.org/stable/2408753>.
- M. Manhart and A. V. Morozov. Protein folding and binding can emerge as evolutionary spandrels through structural coupling. *Proc Natl Acad Sci*, 112(6):1797–1802, 2015. doi: 10.1073/pnas.1415895112.
- J. H. McDonald and M. Kreitman. Adaptive protein evolution at the Adh locus in Drosophila. *Nature*, 351(6328):652–654, 1991.
- H. J. Muller. The relation of recombination to mutational advance. *Mutat Res*, 106:2–9, 1964. doi: 10.1016/0027-5107(64)90047-8.
- R. A. Neher. Genetic draft, selective interference, and population genetics of rapid adaptation. *Annu Rev Ecol Evol Syst*, 44(1):195–215, 2013. doi: 10.1146/annurev-ecolsys-110512-135920. URL <http://dx.doi.org/10.1146/annurev-ecolsys-110512-135920>.
- R. A. Neher and O. Hallatschek. Genealogies of rapidly adapting populations. *Proc Natl Acad Sci*, 110(2):437–442, 2013. doi: 10.1073/pnas.1213113110. URL <http://www.pnas.org/content/110/2/437.abstract>.
- R. A. Neher and B. I. Shraiman. Competition between recombination and epistasis can cause a transition from allele to genotype selection. *Proc Natl Acad Sci U S A*, 106(16):6866–6871, Apr. 2009.
- R. A. Neher, T. A. Kessinger, and B. I. Shraiman. Coalescence and genetic diversity in sexual populations under selection. *Proc Natl Acad Sci*, 110(39):15836–15841, 2013. doi: 10.1073/pnas.1309697110. URL <http://www.pnas.org/content/110/39/15836.abstract>.
- A. Nourmohammad, S. Schiffels, and M. Lässig. Evolution of molecular phenotypes under stabilizing selection. *J Stat Mech Theor Exp*, 2013(01):P01012, Jan. 2013. doi: 10.1088/1742-5468/2013/01/P01012. URL <http://stacks.iop.org/1742-5468/2013/i=01/a=P01012?key=crossref.cf2d47909128801e1522f4247cbd4be2>.
- A. Nourmohammad, J. Rambeau, T. Held, V. Kovacova, J. Berg, and M. Lässig. Adaptive evolution of gene expression in drosophila. *Cell reports*, 20(6):1385–1395, 2017. doi: 10.1016/j.celrep.2017.07.033.
- H. A. Orr. The genetic theory of adaptation: a brief history. *Nature Rev. Genet.*, 6(2):119–127, 2005. doi: 10.1038/nrg1523.

- D. P. Rice, B. H. Good, and M. M. Desai. The evolutionarily stable distribution of fitness effects. *Genetics*, 200(1):321–329, 2015. doi: 10.1534/genetics.114.173815. URL <http://www.genetics.org/content/200/1/321.abstract>.
- I. M. Rouzine, É. Brunet, and C. O. Wilke. The traveling-wave approach to asexual evolution: Muller’s ratchet and speed of adaptation. *Theor Popul Biol*, 73(1):24 – 46, 2008. ISSN 0040-5809. doi: 10.1016/j.tpb.2007.10.004. URL <http://www.sciencedirect.com/science/article/pii/S0040580907001207>.
- S. Schiffels, G. J. Szöllösi, V. Mustonen, and M. Lässig. Emergent neutrality in adaptive asexual evolution. *Genetics*, 189(4):1361–1375, 2011. doi: 10.1534/genetics.111.132027. URL <http://www.genetics.org/content/189/4/1361.full.pdf>.
- A. W. Serohijos and E. I. Shakhnovich. Merging molecular mechanism and evolution: theory and computation at the interface of biophysics and evolutionary population genetics. *Curr Opin Struct Biol*, 26(0): 84–91, 2014. ISSN 0959-440X. doi: 10.1016/j.sbi.2014.05.005. URL <http://www.sciencedirect.com/science/article/pii/S0959440X14000591>.
- O. Tenaillon, J. E. Barrick, N. Ribeck, D. E. Deatherage, J. L. Blanchard, A. Dasgupta, G. C. Wu, S. Wielgoss, S. Cruveiller, C. Médigue, D. Schneider, and R. E. Lenski. Tempo and mode of genome evolution in a 50,000-generation experiment. *Nature*, 536:165 EP –, 08 2016. URL <http://dx.doi.org/10.1038/nature18959>.
- L. S. Tsimring, H. Levine, and D. A. Kessler. Rna virus evolution via a fitness-space model. *Phys Rev Lett*, 76:4440–4443, Jun 1996. doi: 10.1103/PhysRevLett.76.4440. URL <https://link.aps.org/doi/10.1103/PhysRevLett.76.4440>.
- D. B. Weissman and N. H. Barton. Limits to the rate of adaptive substitution in sexual populations. *PLos Genet*, 8(6):1–18, 06 2012. doi: 10.1371/journal.pgen.1002740. URL <http://dx.doi.org/10.1371/journal.pgen.1002740>.
- D. B. Weissman and O. Hallatschek. The rate of adaptation in large sexual populations with linear chromosomes. *Genetics*, 196(4):1167–1183, 2014. ISSN 0016-6731. doi: 10.1534/genetics.113.160705. URL <http://www.genetics.org/content/196/4/1167>.
- K. B. Zeldovich, P. Chen, and E. I. Shakhnovich. Protein stability imposes limits on organism complexity and speed of molecular evolution. *Proc Natl Acad Sci*, 104(41):16152–16157, 2007. doi: 10.1073/pnas.0705366104. URL <http://www.pnas.org/content/104/41/16152.abstract>.

Reviewers' comments:

Reviewer #1 (Remarks to the Author):

The authors have adequately addressed my concerns, primarily by improving the readability of the paper.

Reviewer #2 (Remarks to the Author):

I appreciate the authors' responses to my questions and comments. They have made some useful improvements to the paper, but as it stands I think it is still nearly impenetrable to all but specialists who have carefully studied the previous literature on fitness waves and clonal interference. I was unable to follow the steps to the vast majority of mathematical results, even while carefully going through the Methods and SI. I think the results are interesting, but the paper still needs a substantial revision of its presentation to be suitable for publication.

General comments:

1. I think the paper would hugely benefit from writing a self-contained, bottom-up derivation of all the results in the SI, so that a dedicated reader can see how to start from the basic model and its parameters and then systematically derive all the results in a linear fashion. As it stands, I found myself trying to piece together all the definitions and steps from the main text, Methods, and SI, but I quickly got lost every time.
2. Another major help would be a table for all the notation. (Perhaps an abbreviated version in the main text, and a complete version in the SI.) Besides defining each term (both mathematically and conceptually), the table should categorize whether the quantity is a property of individual genes (local) or a property of the entire genome (global), and whether it is an input parameter or an output of the dynamics. Indeed, since those distinctions are a central theme of this work, organizing the table this way would further support those themes.
3. Along these same lines, I had a hard time distinguishing between definitions and derived relations between quantities throughout the paper. This was compounded by the authors' use of chaining together multiple relations in almost every equation ($a = b = c$). I think these multi-part equations should be used sparingly, or not at all, in the main text.

Specific comments:

1. I would suggest avoiding the words "coexist" and "coexisting" in Box 1 since to ecologists these words imply there is stabilizing selection that maintains multiple variants for arbitrarily long times, rather than multiple variants that happen to be present in the same population at the same time (but that will quickly change). Maybe just say "multiple variants exist" or "multiple variants are present" in the population.
2. As an example of my general comments above, I still do not understand where Eqs. 1 and 2 come from. The main text says they are derived in the Methods, but the derivation was too terse or incomplete for me to follow. Equation 1 apparently comes from Eq. 17, which comes from Eq. 16, but where does this equation come from? What is \mathcal{L}_Δ ? There are quite a lot of steps here that seem glossed over.
3. Where is the exact definition of Δ_G ? It just appears in Eq. 15 without any real explanation. Furthermore, this section still did not explain what the quantity Δ_G

$f'(\Gamma)$ is, as I mentioned in my earlier review. I understand conceptually that it is the contribution to the increase of Γ from selection, but this needs to be explained mathematically.

4. Equation 15 also assumes that the supply of mutations driving a protein toward lower stability is constant, but this is not true: this supply is a function of the stability G , since the more stable the protein is, the more destabilizing mutations there are.

5. Line 493: There is a typo in "mutation-coalescence."

6. Line 493: What does "heritable variance" mean? This phrase sounds like it refers to nongenetic variation in a trait (e.g., stochastic gene expression, which I don't think is part of the paper), but I thought Δ_G was the variance in a particular G across the population.

7. The coalescence rate is a central concept in the results, but I can find neither a precise definition nor a conceptual explanation of what the authors mean by it in the context of their model.

8. I did not understand the derivation of Eq. 8. The authors say that for a stable wave, mutation rate must exceed average selection, so $u > s$. But if I substitute in Eq. 6 ($s = 4ug/c$) as the authors say, I get $u > 4ug/c$, which leads to a condition on c ($c > 4$) not g , since the g drops out.

9. Lines 234-235: The previous sentences indicate the mutational load \mathcal{L} for an individual gene can be approximated as $f_0 - f(\Gamma) \approx f_0 e^{-\Gamma/kT}$. But then this sentence says I should use Eq. 6 --- how? That equation does not have any relation for Γ .

10. Lines 284-285 assert that the physiological mutational load is similar to the interference load using *E. coli* parameters, but don't the numbers show the interference load is much larger? The estimate for the interference load in microbes is $3e-2$ (line 267), but the physiological load estimate seems to be $1/5000 \sim 2e-4$ (lines 283-284).

11. Lines 294-295 mention the "stability condition $f'(G)$ " --- shouldn't there be an inequality here?

12. Lines 307-308 and Eq. 11: What is the logic from "each additional deleterious trait changes are only marginally selected" to "the relaxation time is of the order of the inverse mutation rate"?

13. Figure 3 caption: I think the two parts of panel (a) should be marked "top" and "bottom," not "left" and "right."

14. Line 347 includes the relation $R/\tilde{\sigma}(\xi)$. Is this a definition of ξ ? Or an independent constraint on R and ξ ?

15. Line 354 mentions that there is a first-order phase transition from genotypic selection at low recombination rates to allelic selection at high recombination rates, but I didn't see any explanation or evidence for this being a sharp transition.

16. I'm still skeptical about the mathematical formulation of the fitness function of protein stabilities in Eqs. 13 and 14, especially as it applies to essential genes. The authors say it can account for essential proteins by taking $f_0 \rightarrow \infty$, but that will make such a protein always lethal for any finite stability. (Also, it seems like an undesirable parameterization of the model to make essential proteins require some sort of infinite limit.) I think this problem is related to their choice of adding fitness functions for each protein to get the total fitness of the genotype, since I

would argue sigmoidal fitness functions are more naturally multiplicative. I think a better formulation would be the following. Let $f(G) = (1 + (1 - s)e^{-G/KT}) / (1 + e^{-G/KT})$, so a perfectly-stable protein has maximum fitness 1, and a perfectly-unstable protein has fitness $1 - s$. That is, s is the fitness cost of the unfolded protein, which ranges from 0 (neutral protein) to 1 (essential protein). The fitness for the entire genotype is the product of these fitness functions for each protein. Therefore if an essential protein becomes completely unstable, the fitness for the entire genotype becomes zero.

Reviewer #1

The authors have adequately addressed my concerns, primarily by improving the readability of the paper.

We again thank the reviewer for his valuable criticism that helped us to improve the manuscript.

Reviewer #2

I appreciate the authors' responses to my questions and comments. They have made some useful improvements to the paper, but as it stands I think it is still nearly impenetrable to all but specialists who have carefully studied the previous literature on fitness waves and clonal interference. I was unable to follow the steps to the vast majority of mathematical results, even while carefully going through the Methods and SI. I think the results are interesting, but the paper still needs a substantial revision of its presentation to be suitable for publication.

We again thank the reviewer for comments on the manuscript and detailed feedback on ambiguities. In response, we made changes throughout the manuscript, most substantially in Methods. We reply to the detailed points below.

General comments:

1. I think the paper would hugely benefit from writing a self-contained, bottom-up derivation of all the results in the SI, so that a dedicated reader can see how to start from the basic model and its parameters and then systematically derive all the results in a linear fashion. As it stands, I found myself trying to piece together all the definitions and steps from the main text, Methods, and SI, but I quickly got lost every time.

The self-consistent solution of the coupled global and local genome evolution, which is at the heart of the paper, is admittedly complex. In improving the presentation, we focused on three main points:

- The derivation of the key equations (1) and (2) is expanded in Methods. We have re-organized the presentation to make the derivation more linear, separating steps that build up the solution and assumptions that are checked a posteriori.
- We explain the onset of phenotypic interference, equation (8), in a new paragraph in Methods.
- We extend the discussion of fitness landscapes for essential genes.

2. Another major help would be a table for all the notation. (Perhaps an abbreviated version in the main text, and a complete version in the SI.) Besides defining each term (both mathematically and conceptually), the table should categorize whether the quantity is a property of individual genes (local) or a property of the entire genome (global), and whether it is an input parameter or an output of the dynamics. Indeed, since those distinctions are a central theme of this work, organizing the table this way would further support those themes.

We followed the suggestion and list the relevant variables in Supplementary Table S1.

3. Along these same lines, I had a hard time distinguishing between definitions and derived relations between quantities throughout the paper. This was compounded by the authors' use of chaining together multiple relations in almost every equation ($a = b = c$). I think these multi-part equations should be used sparingly, or not at all, in the main text.

We expanded multiple-step derivations where appropriate and left intermediate steps when they build the argument in a linear way.

Specific comments:

1. I would suggest avoiding the words "coexist" and "coexisting" in Box 1 since to ecologists these words imply there is stabilizing selection that maintains multiple variants for arbitrarily long times, rather than multiple variants that happen to be present in the same population at the same time (but that will quickly change). Maybe just say "multiple variants exist" or "multiple variants are present" in the population.

Thanks for pointing out this ambiguity. We replaced "variants" by "mutations" and "coexisting" by "simultaneously segregating".

2. As an example of my general comments above, I still do not understand where Eqs. 1 and 2 come from. The main text says they are derived in the Methods, but the derivation was too terse or incomplete for me to follow. Equation 1 apparently comes from Eq. 17, which comes from Eq. 16, but where does this equation come from? What is \mathcal{L}_Δ ? There are quite a lot of steps here that seem glossed over.

We carefully reorganized the text in Methods (lines 500 - 545) to improve the derivation:

- In the section "Evolutionary model", we add a new equation (16) describing the evolution of the trait diversity and refer the reader to ref. [37] for a careful derivation.
- In the new section, "Evolutionary equilibria...", we first show that (16) leads to (1) and provide intuition for the solution. We then show that (1) can also be derived from known results on sequence diversity.
- Second, we use the result (1) to derive the statistics of the mean trait, equation (2).
- Third, we justify an approximation made in step 1. This is a self-consistency argument that requires the full closure solution. This self-consistency check is in eqs. (18) and (19), which were (16) and (17) in the previous version. \mathcal{L}_Δ is now formally defined above (18).

3. Where is the exact definition of Δ_G ? It just appears in Eq. 15 without any real explanation.

The diversity Δ_G is formally defined directly above Equation (1); we have added a reference to make it clear that we are using the standard definition of quantitative genetics.

Furthermore, this section still did not explain what the quantity $\Delta_G f'(\Gamma)$ is, as I mentioned in my earlier review. I understand conceptually that it is the contribution to the increase of Γ from selection, but this needs to be explained mathematically.

We give an explicit derivation in equation (17), and the main text refers to this more clearly.

4. Equation 15 also assumes that the supply of mutations driving a protein toward lower stability is constant, but this is not true: this supply is a function of the stability G , since the more stable the protein is, the more destabilizing mutations there are.

We are aware that a full treatment always involves a G -dependent mutational bias; this effect is contained in the simulations and now explicitly mentioned in Methods (line 509). The approximation with a constant $\kappa \sim 1$ is appropriate for functional proteins, where most mutations are deleterious; see, e.g., Zeldovich et al. 2007.

5. Line 493: There is a typo in "mutation-coalescence."

Thanks, we fixed the typo.

6. Line 493: What does "heritable variance" mean? This phrase sounds like it refers to nongenetic variation in a trait (e.g., stochastic gene expression, which I don't think is part of the paper), but I thought Δ_G was the variance in a particular G across the population.

The term refers to the genetic variation, excluding any non-genetic variation (which is not heritable). Δ_G is the variance of the genetically encoded trait variation across the population. The definition of Δ_G above eq. (1) now includes an appropriate reference to the quantitative genetics literature.

7. The coalescence rate is a central concept in the results, but I can find neither a precise definition nor a conceptual explanation of what the authors mean by it in the context of their model.

The coalescence rate is the inverse of the average coalescence time, i.e. the time from the common ancestor of the population. It is the timescale that defines the scale of effective selection [Neher, 2013, Schiffels et al., 2011]. We improved the explanation of this term in the Introduction and in Fig. 1.

8. I did not understand the derivation of Eq. 8. The authors say that for a stable wave, mutation rate must exceed average selection, so $ug > s$. But if I substitute in Eq. 6 ($s = 4ug/c$) as the authors say, I get $ug > 4ug/c$, which leads to a condition on c ($c > 4$) not g , since the g drops out.

Thanks, we corrected this inconsistency and added a straightforward derivation of the onset threshold in Methods (lines 580 - 585).

9. Lines 234-235: The previous sentences indicate the mutational load \mathcal{L} for an individual gene can be approximated as $f_0 - f(\Gamma) \approx f_0 e^{-\Gamma/kT}$. But then this sentence says I should use Eq. 6 — how? That equation does not have any relation for Γ .

We improved the text (now lines 239 - 242). The key point is that the product $s = \epsilon_0 f'$ has been evaluated in equation (6).

10. Lines 284-285 assert that the physiological mutational load is similar to the interference load using *E. coli* parameters, but don't the numbers show the interference load is much larger? The estimate for the interference load in microbes is $3e-2$ (line 267), but the physiological load estimate seems to be $1/5000 \approx 2e-4$ (lines 283-284).

As we specify in line 266, the cost in line 267 is for a 10% more genes, i.e. for 500 additional genes. Therefore, the costs per gene are of the same order of magnitude.

11. Lines 294-295 mention the "stability condition $f'(G)$ " — shouldn't there be an inequality here?

We added a reference to the detailed balance condition, Equation (2). Moreover, we specified the discussion of (in-)stability regimes below with respect to f_0 , which clarifies the condition and the equation (10).

12. Lines 307-308 and Eq. 11: What is the logic from "each additional deleterious trait changes are only marginally selected" to "the relaxation time is of the order of the inverse mutation rate"?

The logic is, marginally selected mutations evolve in a near-neutral way. Therefore, the fixation rate of these mutations becomes similar to the mutation rate. We specified the intermediate argument in the text.

13. Figure 3 caption: I think the two parts of panel (a) should be marked "top" and "bottom," not "left" and "right."

Thanks, we fixed this.

14. Line 347 includes the relation $R\xi/g\bar{\sigma}(\xi)$. Is this a definition of ξ ? Or an independent constraint on R and ξ ?

This relation is discussed in detail in refs. [Neher, 2013, Neher et al., 2013, Weissman and Barton, 2012, Weissman and Hallatschek, 2014]. It is a self-consistent relation between block size and recombination following from simple time-scale arguments, which is now specified in the text.

15. Line 354 mentions that there is a first-order phase transition from genotypic selection at low recombination rates to allelic selection at high recombination rates, but I didn't see any explanation or evidence for this being a sharp transition.

The evidence comes from analytical considerations in the section "The pathway to sexual evolution" predicting the first order transition. We improved the discussion of these analytical results. In particular, in a finite system, first order transitions are always rounded. In the present system, there are two sources of finite-size scaling: the number of genes g , and the population size N . As expected, the simulation results are rounded consistently for different gene numbers. We now comment on this directly in Fig. 5a.

16. I'm still skeptical about the mathematical formulation of the fitness function of protein stabilities in Eqs. 13 and 14, especially as it applies to essential genes. The authors say it can account for essential proteins by taking $f_0 \rightarrow \infty$, but that will make such a protein always lethal for any finite stability. (Also, it seems like an undesirable parameterization of the model to make essential proteins require some sort of infinite limit.)

We improved the text in the Methods section "Biophysical fitness models" to make it clear that we do not model essential genes by $f_0 \rightarrow \infty$. Rather, zero growth (Wright fitness $w = 0$, lethality) occurring at a finite

threshold free energy G_0 translates into a singularity of the Malthusian (log) fitness, $f(G) = \log w(G) \rightarrow -\infty$ for $G \rightarrow G_0$ at finite f_0 . This is similar to the modeling of essential genes in [Chen and Shakhnovich, 2009, Zeldovich et al., 2007]. Our simulations with essential genes are done with this well-defined fitness.

I think this problem is related to their choice of adding fitness functions for each protein to get the total fitness of the genotype, since I would argue sigmoidal fitness functions are more naturally multiplicative. I think a better formulation would be the following. Let $f(G) = (1 + (1 - s)e^{-G/kT}) / (1 + e^{-G/kT})$, so a perfectly-stable protein has maximum fitness 1, and a perfectly-unstable protein has fitness $1 - s$. That is, s is the fitness cost of the unfolded protein, which ranges from 0 (neutral protein) to 1 (essential protein). The fitness for the entire genotype is the product of these fitness functions for each protein. Therefore if an essential protein becomes completely unstable, the fitness for the entire genotype becomes zero.

Thanks for pointing to this interesting model of selection on essential genes, which we included in the section “Biophysical fitness models”, translating it into the language of Malthusian (log) fitness. We then recall a salient point of our analysis: the mutation-selection balance drives the average load per gene to a fixed value $2\bar{\sigma}$ independently of the gene selection amplitude f_0 . That is, the equilibrium load of essential genes is no different from others, the larger f_0 is compensated by a larger equilibrium G . Therefore, the model assumption (14) of multiplicative fitness across genes, i.e., additive Malthusian (log) fitness, is consistent in this regime.

References

- P. Chen and E. I. Shakhnovich. Lethal mutagenesis in viruses and bacteria. *Genetics*, 183(2):639–650, 2009. doi: 10.1534/genetics.109.106492. URL <http://www.genetics.org/content/183/2/639.abstract>.
- R. A. Fisher. *The genetical theory of natural selection*. Oxford Clarendon Press, 1930. doi: 10.5962/bhl.title.27468.
- R. A. Neher. Genetic draft, selective interference, and population genetics of rapid adaptation. *Annu Rev Ecol Evol Syst*, 44(1):195–215, 2013. doi: 10.1146/annurev-ecolsys-110512-135920. URL <http://dx.doi.org/10.1146/annurev-ecolsys-110512-135920>.
- R. A. Neher, T. A. Kessinger, and B. I. Shraiman. Coalescence and genetic diversity in sexual populations under selection. *Proc Natl Acad Sci*, 110(39):15836–15841, 2013. doi: 10.1073/pnas.1309697110. URL <http://www.pnas.org/content/110/39/15836.abstract>.
- S. Schiffels, G. J. Szöllösi, V. Mustonen, and M. Lässig. Emergent neutrality in adaptive asexual evolution. *Genetics*, 189(4):1361–1375, 2011. doi: 10.1534/genetics.111.132027. URL <http://www.genetics.org/content/189/4/1361.full.pdf>.
- D. B. Weissman and N. H. Barton. Limits to the rate of adaptive substitution in sexual populations. *PLoS Genet*, 8(6):1–18, 06 2012. doi: 10.1371/journal.pgen.1002740. URL <http://dx.doi.org/10.1371/journal.pgen.1002740>.
- D. B. Weissman and O. Hallatschek. The rate of adaptation in large sexual populations with linear chromosomes. *Genetics*, 196(4):1167–1183, 2014. ISSN 0016-6731. doi: 10.1534/genetics.113.160705. URL <http://www.genetics.org/content/196/4/1167>.
- K. B. Zeldovich, P. Chen, and E. I. Shakhnovich. Protein stability imposes limits on organism complexity and speed of molecular evolution. *Proc Natl Acad Sci*, 104(41):16152–16157, 2007. doi: 10.1073/pnas.0705366104. URL <http://www.pnas.org/content/104/41/16152.abstract>.

REVIEWERS' COMMENTS:

Reviewer #2 (Remarks to the Author):

The authors have made a few more improvements to the text, but they have not implemented the substantial revision that I think this paper needs to be publishable in a general-readership journal like Nature Communications. There are only very minor changes to the main text results on the fitness wave of quantitative traits (Eqs. 1-8), leaving it much too dense still with technical details to be accessible to anyone without substantial familiarity with both the quantitative trait and fitness wave literature. Furthermore, there is still no self-contained, linear derivation of all the results in the SI, as I requested after the previous revision. I think these sections need to be completely rewritten. Otherwise, the paper would better be published in a specialty journal.